# The endoribonuclease Arlr is required to maintain lipid homeostasis by downregulating lipolytic genes during aging

Xiaowei Sun[1], Jie Shen [1], Norbert Perrimon [2,3], Xue Kong [1] & Dan Wang [1]✉

While disorders in lipid metabolism have been associated with aging and age-related diseases, how lipid metabolism is regulated during aging is poorly understood. Here, we characterize the *Drosophila* endoribonuclease CG2145, an ortholog of mammalian EndoU that we named Age-related lipid regulator (Arlr), as a regulator of lipid homeostasis during aging. In adult adipose tissues, Arlr is necessary for maintenance of lipid storage in lipid droplets (LDs) as flies age, a phenotype that can be rescued by either high-fat or high-glucose diet. Interestingly, RNA-seq of *arlr* mutant adipose tissues and RIP-seq suggest that Arlr affects lipid metabolism through the degradation of the mRNAs of lipolysis genes – a model further supported by the observation that knockdown of *Lsd-1, regucalcin, yip2* or *CG5162*, which encode genes involved in lipolysis, rescue the LD defects of *arlr* mutants. In addition, we characterize DendoU as a functional paralog of Arlr and show that human ENDOU can rescue *arlr* mutants. Altogether, our study reveals a role of ENDOU-like endonucleases as negative regulator of lipolysis.

Lipid droplets (LDs) are highly dynamic organelles that contain a core of neutral lipids including triacylglycerols (TAGs) and cholesteryl esters, which are surrounded by a phospholipid monolayer[1,2]. While mostly found in the adipose tissue, LDs are also present in other cells including hepatocytes, enterocytes, macrophages, and adrenocortical cells to store excess lipids. These spherical organelles are regulated by LD-associated proteins which regulate LD biogenesis and degradation, and that play a crucial role in maintaining lipid homeostasis and energy supply[3].

LDs originate from the bilayer endoplasmic reticulum (ER)[4–7] where the curvature of ER tubules (smooth ER) catalyzes the nucleation of neutral lipids into lenses leading to nascent LD buds[8,9]. During de novo synthesis, LD size is regulated by Seipin and the Fat Storage-inducing Transmembrane (FIT) proteins, which are associated with lens formation and budding[10–13]. In addition, LD growth is regulated by TAG synthesis enzymes (e.g., GPAT4, AGPAT3, DGAT1, and DGAT2), which relocalize from the ER to the LD surface[14]. An alternative mechanism of LD growth is the

fusion of existing small LDs by either coalescence or lipid exchange[15–17].

LD degradation is mediated by lipolysis and lipophagy pathways. During lipolysis, TAG hydrolysis is initiated by adipose triglyceride lipase (ATGL), followed by hormone-sensitive lipase (HSL) that hydrolyzes diacylglycerol (DAG) and monoacylglycerol lipase (MGL) that hydrolyzes monoacylglycerol (MAG), releasing free fatty acids for further degradation through mitochondrial or peroxisomal β-oxidation to generate ATP[18,19]. LDs can also be degraded through autophagy/lipophagy[20] whereby the whole LD is sequestered by autophagosomes, which is then delivered to lysosomes where it is degraded by lipases[20,21]. The lipolysis pathway prefers to hydrolyze larger-sized LDs and produces small LDs, whereas the lipophagy machinery can only engulf small LDs, inhibition of which results in accumulation of small LDs[22].

A number of key proteins have been identified as regulators of the size and number of LDs, such as cell death-inducing DFFA-like effector c (CIDEC) proteins and perilipins[3]. In adipocyte cell lines, FSP27/CIDEC

[1]Department of Plant Biosecurity and MARA Key Laboratory of Surveillance and Management for Plant Quarantine Pests, College of Plant Protection, China Agricultural University, Beijing, China. [2]Department of Genetics, Blavatnik Institute, Harvard Medical School, Boston, MA, USA. [3]Howard Hughes Medical Institute, Boston, MA, USA. ✉e-mail: dwang@cau.edu.cn

is enriched at LD-LD contact sites and mediates directional transfer of lipids from smaller to larger LDs[17]. Located at the surface of LDs, PLIN1, the best characterized LD perilipin (PLIN), plays a dual role in lipid metabolism, limiting lipase access to stored TAGs in the fed state or facilitating in the fast state hormonally-stimulated lipolysis[23–25]. Phosphorylated PLIN1 serves as a scaffold for lipases to drive lipolysis[21,23,24], and for proteins such as FSP27 to regulate LD growth[3]. Consistent with this, in *Drosophila* the lipid storage droplet protein 1 (Lsd-1/PLIN1) stimulates lipolysis by recruiting HSL to the LD surface[26–28], while Lsd-2/PLIN2 promotes lipogenesis by antagonizing ATGL/Brummer (Bmm) activity[29,30].

Understanding LD dynamics and metabolism is of great significance for lipid metabolism-associated disorders, aging, and lifespan[18,31,32]. For example, during aging a change in overall lipid metabolism has been observed, usually leading to lipid accumulation in the adipose tissue and ectopic accumulation in non-adipose tissues, which increases the risk of developing metabolic diseases[33]. Other studies have demonstrated that the activity of HSL is reduced in the aged adipose tissue[34,35], which is consistent with lipid changes in aging organisms. Further, genetic studies on enzymes that regulate biosynthesis and mobilization of neutral lipids, such as diacylglycerol acyltransferase 1 (DGAT1) in mice[36], or ATGL and diacylglycerol lipase (DAGL) in *C. elegans* and *Drosophila*[37–40], support the association of increased lipolysis with longevity. On the other hand, increased TAG levels with extended lifespan are also observed along with increased fat synthesis and breakdown[41,42], indicating that increased lipid mobilization helps extend lifespan[42,43]. In addition, lipid metabolism is regulated by highly conserved pro-longevity signaling pathways, such as insulin/insulin-like growth factor signaling (IIS) and mechanistic target of rapamycin (mTOR) signaling, as well as by dietary restriction. These pathways extend lifespan, usually through activation of lipolysis via upregulation of lipases, desaturation of fatty acids, and lipophagy[18]. In particular, the transcription factor FoxO, which is inhibited by IIS and activated by starvation, stimulates ATGL/Bmm and other lipases through transcription or indirect modifications to promote lipolysis[44–46]. Despite these studies, how LDs are regulated during aging is poorly understood.

Here, we identify the endoribonuclease (EndoU) Arlr as an age-related lipid regulator. The EndoU family proteins have a conserved RNA binding domain that cleaves single-stranded RNA harboring U-rich sequences[47–52]. We find that high expression levels of Arlr during aging are essential for lipid accumulation in LDs and that loss of *arlr* results in rapid lipid consumption. Mechanistically, we show that Arlr controls the homeostasis of LDs by affecting the stability of mRNAs encoding proteins involved in lipolysis. In addition, we demonstrate that both a paralog of Arlr, DendoU, and human ENDOU also act as negative regulators of lipolysis, as they can recue *arlr* mutants.

## Results

### Lipid storage is reduced in *arlr* mutants during aging

To investigate the function of *arlr* in vivo, we analyzed the expression levels of *arlr* at different developmental stages. Interestingly, although *arlr* is expressed at all stages, expression of *arlr* was much higher during adult stages, which is consistent with a previous report[50]. Further, *arlr* mRNA expression was high at all adult stages examined (1 week to 5 weeks) (Fig. 1a). Consistent with this, examination of single nuclei RNAseq (snRNAseq) dataset from the Fly Cell Atlas (FCA) and Aging Fly Cell Atlas (AFCA) reveal higher expression levels of *arlr* in fat body nuclei at 30, 50, and 70 days as compared with 5 days (Supplementary Fig. 1a)[53,54]. FCA data also indicate that among the 15 groups of individual tissues, *arlr* is only highly expressed in the fat body[53]. In addition, we confirmed that Arlr was expressed in the fat body, colocalizing with the ER. The endogenous Arlr-GFP fusion protein showed a punctate distribution in the cytoplasm that co-localized with KDEL-mCherry, a marker of the ER (*arlr-GFP; ppl > KDEL-mCherry*). Further,

an HA-tagged form of Arlr (Arlr-HA) also co-localized with the ER marker (Supplementary Fig. 2a, b). These observations are consistent with our previous proximity labeling analysis where we identified Arlr as a fat body-expressed protein[55]. Altogether, these results suggest that Arlr plays an important role in the fat body during aging.

Next, to analyze the role of *arlr* in adipose tissues, we generated two null mutant alleles, *arlr^262^* and *arlr^364^*, using CRISPR/Cas9 mediated genome editing. *arlr^262^* and *arlr^364^* were associated with a deletion of 262 bp and 364 bp in the coding sequence region, respectively (Supplementary Fig. 3a, b). Interestingly, although no mRNA was detected with primers flanking the deleted regions associated with the two mutations, we did observe residual mRNA expression with a primer pair located at the N-terminus (Supplementary Fig. 3c), indicating that these mutations lead to truncated mRNAs. Homozygous mutants are viable, indicating that loss of *arlr* is not essential for zygotic viability. Whereas larvae and young (1 week old) *arlr^262^* mutant adults showed no obvious phenotypes in body size and lipid storage (Supplementary Fig. 4), which is consistent with our previous study[55], 3-week-old *arlr* mutants (*arlr^262^* and *arlr^364^*) exhibited smaller LDs in both females (Fig. 1b, Supplementary Fig. 5a–c) and males (Supplementary Fig. 6b, c). This phenotype was even more obvious in 5-week-old flies (Fig. 1b, Supplementary Fig. 6e, f). Quantification of LD sizes showed a greater reduction of large droplets and an increase in small droplets (Fig. 1c, d, Supplementary Fig. 6g, h). The proportion of LDs measuring less than 4.0 μm² accounted for 86.5% and 77.7% of the total in *arlr^262^* and *arlr^364^* mutants, respectively, compared to 52.5% in control flies (Fig. 1d′). In addition, LDs larger than 14.0 μm² accounted for only 1.1% and 1.6% of the total in *arlr^262^* and *arlr^364^* mutants, respectively, compared to 18.3% in control flies. Further, TAG levels, an indicator of lipid storage, were slightly decreased in 3-week-old flies and greatly reduced in 5-week-old flies (Fig. 1e, Supplementary Fig. 6i). Consistent with the mutant phenotypes, interfering with *arlr* expression in the fat body using two independent RNAi lines resulted in similar defects (Supplementary Fig. 7). In contrast to the effect on LDs, there were no obvious effects on the levels of glucose and trehalose in *arlr* mutants (Fig. 1f, g, Supplementary Fig. 6j, k). Altogether, these results suggest that Arlr regulates lipid metabolism during aging, resulting in a reduction in LD sizes.

Since LDs and TAG levels are reduced in older *arlr* mutant flies, we wondered whether Arlr affects lifespan. Interestingly, *arlr* mutant females (*arlr^262^* or *arlr^364^*) exhibited shorter lifespan (about 80 days) compared to the control (about 100 days) (Fig. 1h). The median lifespan was 60 and 65 days in the mutants, while it was 80 days in the control. Male flies showed a similar trend, with a 45-days lifespan in *arlr* mutants compared to 60 days in control animals (Supplementary Fig. 6l). Again, lifespan was reduced in *arlr*-knockdown flies (Supplementary Fig. 7d, h). Thus, loss of *arlr* decreases lifespan.

### The EndoU-like domain is necessary for the role of Arlr in lipid metabolism

Arlr is a 592 amino acids (a.a.) protein with four distinct domains, a putative signal peptide, a proline-rich domain, a glycine-rich domain, and an EndoU-like domain (Supplementary Fig. 3a). We generated several transgenic flies to define the region(s) essential for LD storage, and validated them by PCR, qRT-PCR, and Western blots (Supplementary Fig. 3). Expressing full-length tagged Arlr in the fat body (*ppl > UAS-arlr-HA*) rescued the LD storage defects, i.e., lipid size and TAG levels (Fig. 2e, j, k). Interestingly, flies expressing the protein without the signal peptide (SP) (*ppl > UAS-arlr^ΔSP^-HA*) showed larger LDs (Fig. 2f, j, k). This gain-of-function phenotype suggests that the function of Arlr in the fat body is independent of its putative signal peptide and does not require secretion. We also constructed a tagged transgenic fly lacking the conserved EndoU-like domain (*UAS-arlr^N^-HA*). Expression of *arlr^N^-HA* in the fat body (*ppl > UAS-arlr^N^-HA*) of *arlr* mutants could not rescue the LD defects or TAG levels (Fig. 2i–k),

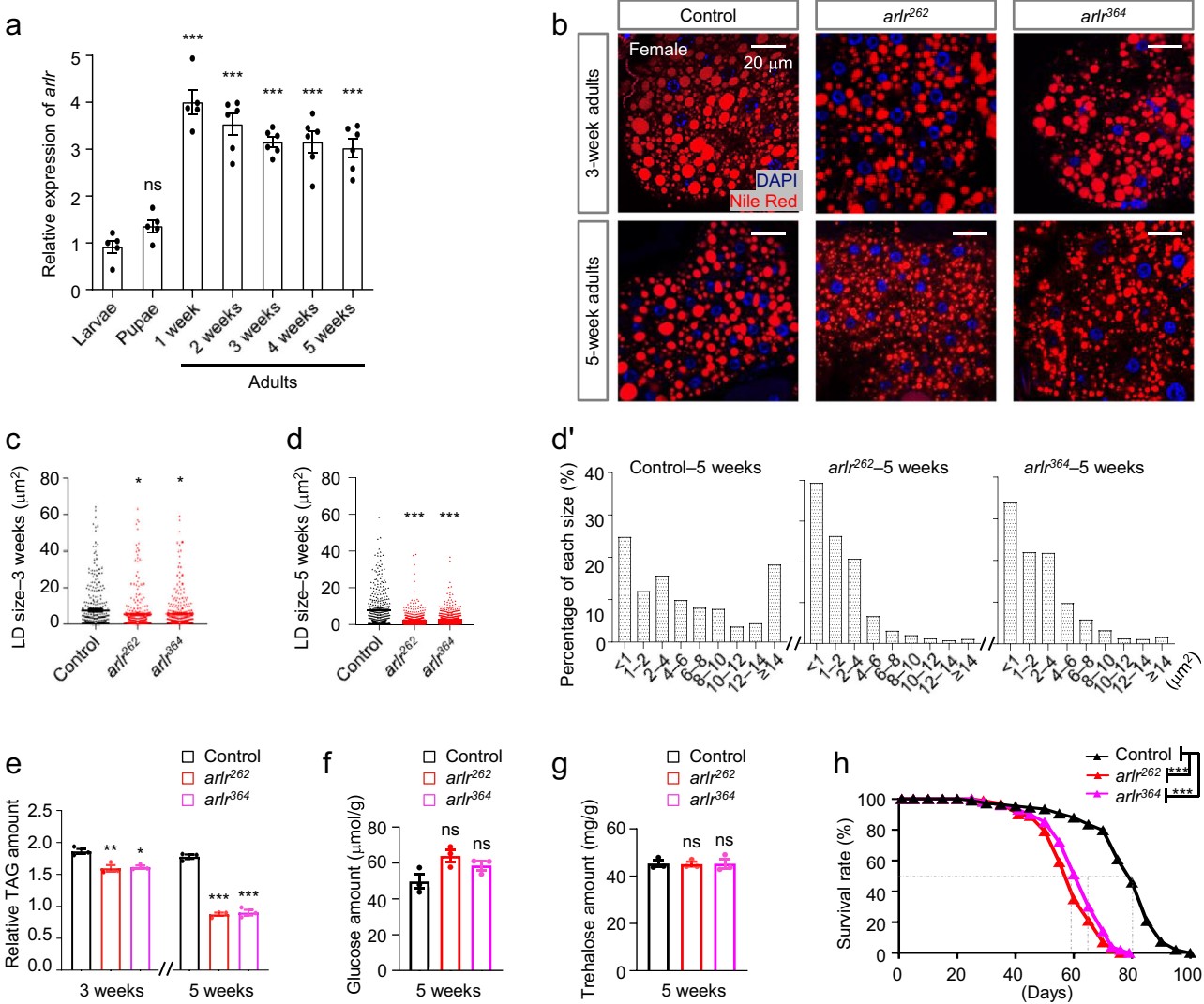

**Fig. 1 | *arlr* mutants have reduced lipid storage.** All panels in this Figure are from females. **a** Relative expression levels of *arlr* at various developmental times. *n* = 5 or 6 biologically independent experiments. ns, no significant difference with the control, *P* = 0.7; \*\*\**P* < 0.001 from 1 week to 5 weeks. **b** Lipid staining illustrating the increase of small LDs in aged *arlr* mutant flies. LDs (Nile Red, red) shown are adjacent to the nucleus (DAPI, blue) at the tissue mid-plane unless mentioned elsewhere. 3-week-old *arlr* mutants (*arlr^262^* and *arlr^364^*) have smaller LDs, a phenotype that is more severe in 5-week-old flies. **c**, **d** Reduction in average LD area in *arlr* mutants. The percentage of large LDs (>14 μm²) was reduced, while the small LDs (<4.0 μm²) was significantly increased (**d'**). For (**d'**), we sorted all LDs by size for each genotype and counted their numbers which were then divided to the total LD number. *n* = 6 biologically independent animals (3 samples in each animal). \**P* = 0.02 for *arlr^262^* and \**P* = 0.04 for *arlr^364^* in (**c**); \*\*\**P* < 0.001 in (**d**). **e** Relative reduction in TAG amount in *arlr* mutant flies. \*\**P* = 0.007 and \**P* = 0.01 at 3 weeks; \*\*\**P* < 0.001 at 5 weeks. **f**, **g** Glucose and trehalose amounts of the whole body were normal. *n* = 3 biologically independent experiments in (**e**–**g**). ns *P* = 0.0535 for *arlr^262^* and *P* = 0.2254 for *arlr^364^* in (**f**); ns *P* > 0.99 in (**g**). **h** Lifespan of female flies. Dashed lines indicate the median lifespan. *n* = 1 biologically independent experiment. \*\*\**P* < 0.001. Statistical data were analyzed by one-way ANOVA with Tukey's multiple comparison test in (**a**) and (**c**–**g**), and by Log-rank (Mantel-Cox) test in (**h**). Error bars represent SEM. Scale bars in (**b**) are 20 μm.

indicating that the EndoU-like domain is necessary for Arlr function in LDs.

### *arlr* mutants show an increase in lipid consumption

Fat accumulation is determined by the balance between fat synthesis and fat breakdown. Under starvation conditions, stored TAGs are hydrolyzed into fatty acids and glycerol to meet the energy demand. To investigate the role of Arlr in lipid homeostasis, we starved 1-week-old flies to activate lipolysis. When starved, control flies consumed lipids, as evidenced by a reduction in the number of LDs after 24 h of starvation. Strikingly, after 24 h of starvation, *arlr^262^* mutants contained fewer LDs, especially large LDs (Fig. 3a, c), suggesting a faster consumption of lipids. In addition, the TAG level of *arlr* mutants was about 70% less than control (Fig. 3d). 5-week mutants starved for 24 h also

showed faster lipid consumption (data not shown). Altogether these results suggest that lipolysis is increased in the absence of Arlr.

Next, to investigate whether lipid synthesis is regulated by Arlr, we fed 5-week-old flies for a period of 24 h with either a high sugar or a high-fat diet (see "Methods"). Under these conditions, LD sizes and numbers in *arlr* mutant fat bodies were similar to controls (Fig. 3b, c) and TAG levels were restored (Fig. 3d). These results indicate that the amount of fat can be replenished through food uptake.

Next, we tested whether the decrease in lifespan of *arlr* mutants is related to reduced nutrition by feeding flies with either a low or a high nutrition diet. *arlr^262^* mutant female flies fed with a poor nutrition diet had a shorter lifespan (40 days compared to 50 days for the control). The median lifespan was 29 days and 40 days, respectively. In contrast, the lifespan was similar to the control (about 100 days, Fig. 3e) when

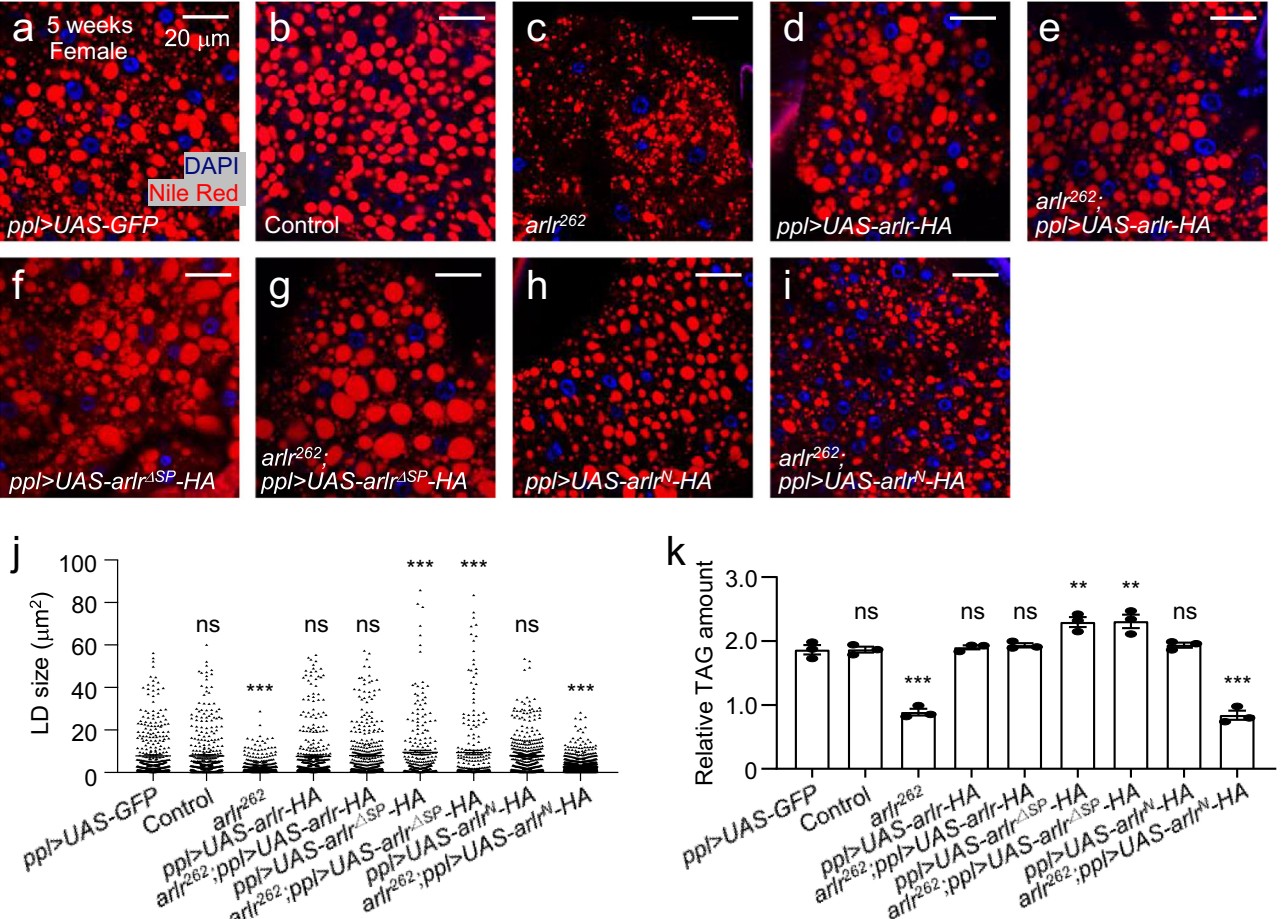

**Fig. 2 | The EndoU-like domain is required for the rescue of lipid defects.** All panels in this Figure are from females. **a, b** LDs in control flies (*ppl > UAS-GFP* and control). **c** 5-week-old *arlr²⁶²* mutant flies with small LDs. **d** Ectopic expression of full-length *arlr* (*arlr-HA*) in the fat body showed normal LDs. **e** Expression of full-length *arlr* restored LD defects. **f** Ectopic expression of *arlr* that lack the signal peptide (*arlr^ΔSP^-HA*) in the fat body showed larger LDs. **g** Expression of *arlr^ΔSP^-HA* restored LD defects in *arlr* mutants. **h, i** *arlr* lacking the EndoU-like domain (*arlr^N^-HA*) failed to restore the lipid defects. **j** Quantification of the rescue phenotype of LD size following expression of *arlr* in *arlr* mutants. *n* = 6 biologically independent animals. Data were analyzed by Mixed-effects analysis Dunnett's multiple comparison test (two-tailed). ns, no significant difference with the *ppl > UAS-GFP* control, *P* = 0.68 for Control, *P* = 0.98 for *ppl > UAS-arlr-HA*, *P* = 0.25 for *arlr²⁶²;ppl > UAS-arlr-HA*, and *P* = 0.3 for *ppl > UAS-arlr^N^-HA*; ***P* < 0.001. **k** Quantification of the TAG rescue following expression of *arlr* in *arlr* mutants. *n* = 3 biologically independent experiments. ns *P* > 0.99; ***P* = 0.003 for *ppl > UAS-arlr^ΔSP-HA^*, ***P* = 0.003 for *arlr²⁶²;ppl > UAS-arlr^ΔSP-HA^*; ****P* < 0.001. Data were analyzed by one-way ANOVA with Tukey's multiple comparison test. Error bars represent SEM. Scale bars in (**a–h**) are 20 μm.

*arlr²⁶²* mutant flies were fed with a high nutrition diet. The median lifespan was 77 days and 81 days, respectively. As lipid consumption was accelerated but lipid replenishment was normal in older mutant flies, this result suggests that lifespan of *arlr* mutants can be reduced by malnutrition and restored by higher nutrition.

### *arlr* genetically interacts with lipolytic genes

Since the EndoU family proteins are endoribonucleases that cleave RNAs with the U-specific recognition sequence[50], we wondered whether Arlr regulates lipolysis by modifying the mRNAs of lipolytic proteins. Thus, we performed RNA-seq to analyze gene expression in *arlr²⁶²* mutant adipose tissues attached to the abdominal cuticle and identified 580 upregulated and 384 downregulated differentially expressed genes (DEGs) (Supplementary Fig. 8a). GO enrichment analysis revealed that the oxidation-reduction process and oxidoreductase activity were the most prominent terms (Supplementary Fig. 8b). Among the DEGs, 48 genes related to lipid metabolism were upregulated and 12 were downregulated (Fig. 4a, Supplementary Fig. 8c, Supplementary Data 1). Strikingly, 40 genes related to lipolysis were within the 60 lipid metabolism-related genes, 10 of which we confirmed by qRT-PCR (Supplementary Fig. 8d). In contrast, few genes

involved in LD biogenesis and growth, as well as autophagy, were found among the DEGs (Supplementary Data 1).

To further explore the contribution of lipolytic genes to the *arlr* mutant phenotype, we tested whether they genetically interact with *arlr*. Four genes including *Lsd-1*, *regucalcin*, *yip2*, and *CG5162* affected the small LD size observed with *arlr²⁶²* (Fig. 4b). Specifically, *arlr²⁶²; ppl>Lsd-1-RNAi* flies showed much larger LDs than *arlr²⁶²* mutants, similar to *ppl>Lsd-1-RNAi* flies[26,27]. In addition, knocking down either *regucalcin*, *yip2*, or *CG5162* in *arlr²⁶²* mutants suppressed the small LD phenotype to normal size (Fig. 4b–d). The TAG amounts were consistent with the rescue of total lipid levels (Fig. 4d). Lsd-1/PLIN1 is one of the major LD surface proteins that stimulates lipolysis[26,27]. *Lsd-1*-overexpressing flies showed a mild LD defect, and expressing *Lsd-1* in the *arlr* mutants did not aggravate the mutant phenotype (Fig. 4c, e), indicating that the small LD defect in *arlr* mutants may depend on the amount of target gene expression and polygenic effects. Regucalcin, also known as Senescence marker protein-30 (SMP30), is a Ca²⁺ binding protein, and in mice hepatocytes SMP30-deficiency show an accumulation of LDs[56]. Yip2 (yippee interacting protein 2) is a homolog of mammalian acetyl-CoA acyltransferase 2 (ACAA2), which catalyzes mitochondrial fatty acid β-oxidation[57]. Finally, CG5162 is a homolog of

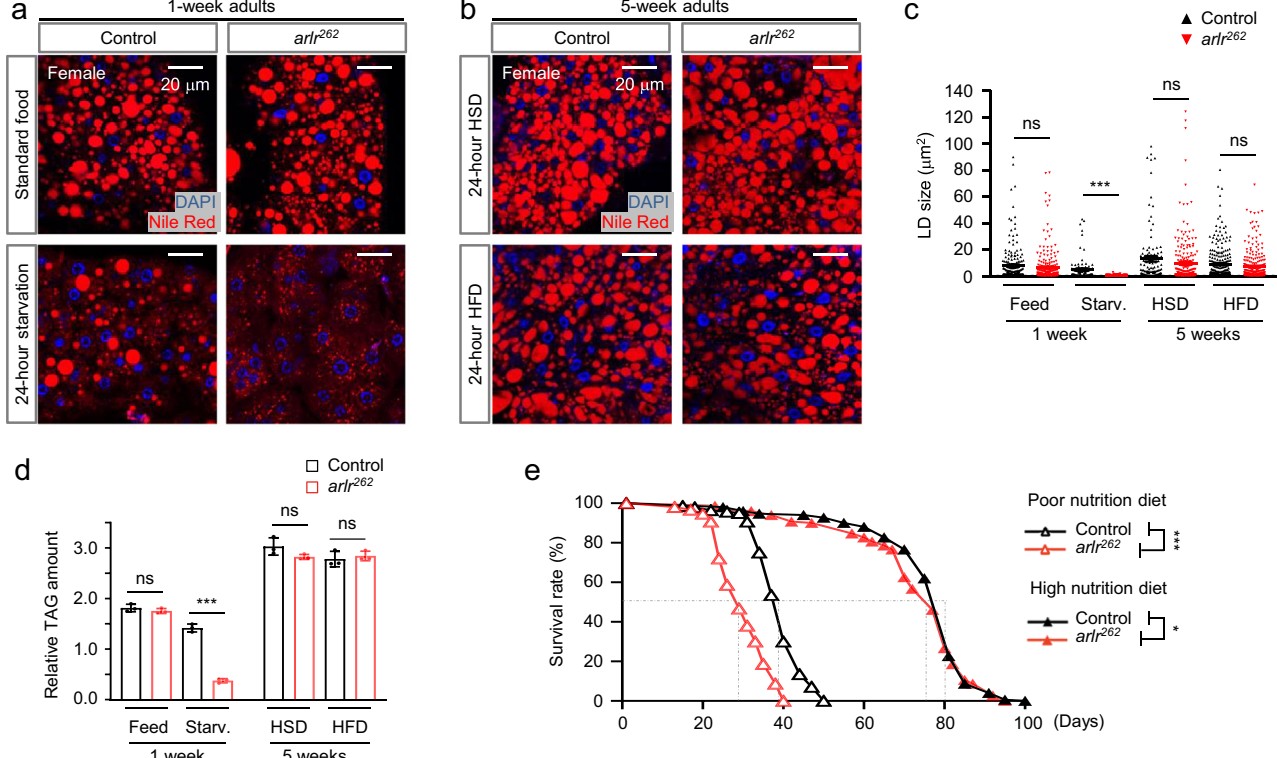

**Fig. 3 | *arlr* mutants show accelerated lipid consumption.** All panels in this Figure are from females. **a** Lipid staining under standard food and starvation conditions. The amount of lipids in *arlr* mutants fed on standard food was comparable to the control at 1 week. After starvation for 24 h, the mutants showed fewer lipids. **b** High sugar diet (HSD) and high-fat diet (HFD) restored lipid levels in aged mutants. 5-week-old *arlr²⁶²* mutant flies fed with HSD or HFD for 24 h showed similar fat storage as control. **c** Quantification of the size of LDs. $n = 6$ biologically independent animals. ns $P = 0.26$ for Feed, $P = 0.14$ for HSD, $P = 0.08$ for HFD; ***$P < 0.001$.

**d** Quantification of the relative TAG amounts. $n = 3$ biologically independent experiments. ns $P = 0.31$ for Feed, $P = 0.12$ for HSD, and $P = 0.59$ for HFD; ***$P < 0.001$. **e** The lifespan of *arlr* mutants was restored by high nutrition diet in females. Dashed lines indicate the median lifespan. $n = 1$ biologically independent experiment. ***$P < 0.001$; *$P = 0.01$. Statistical data were analyzed by independent two-sample $t$ tests (two-tailed) in (**c**) and (**d**), and by Log-rank (Mantel-Cox) test in (**e**). Error bars represent SEM. Scale bars in (**a**) and (**b**) are 20 μm.

---

human lipase C (LIPC)[7,58], which regulates the hydrolysis of triglycerides and phospholipids[59]. These four genes were upregulated in both young and old adults (Supplementary Fig. 8d, e). Thus, among the mRNAs regulated by Arlr are four lipolysis-associated genes involved in lipid metabolism.

### Arlr protein binds to mRNAs encoding lipolytic proteins and negatively regulates their expression levels

To further investigate whether Arlr could directly repress the target genes, we performed luciferase reporter assays. Full-length of Arlr was introduced into the pcDNA3.1 vector to express the Arlr protein. The above four lipolysis-associated genes were introduced into the pmir-GLO vector to quantitatively assess the mRNA activity. Strikingly, the signals of *Lsd-1*, *regucalcin*, *yip2* and *CG5162* were all downregulated in the presence of Arlr (Fig. 5a), indicating that these lipolysis-associated genes are regulated by Arlr.

To explore whether Arlr directly binds to the mRNAs of lipolysis-associated genes, we used GFP-tagged Arlr (Arlr-GFP) flies to immunoprecipitate (IP) mRNAs using antibodies against GFP for cross-linking-based RIP-seq. In the IP group, Arlr showed RNA-binding affinity compared to the input group. Peak distribution analysis showed that 88.9% of the peaks mapped to the coding sequence (CDS) regions, which is significant higher than the expected value. Peaks mapped to mostly 5′ untranslated region (UTR), 3′UTR and intronic regions were lower than the expected value. These data suggest that Arlr preferentially binds to processed mRNAs (Supplementary Fig. 9a). From two independent RIP-seq samples, 2667 genes were found

(Supplementary Fig. 9b). Next, we used the Homer software to analyze the genomic regions corresponding to the peaks and identified the putative binding motifs in the target sequences containing the cleavage sites between AC, AU, UC, and UU[50] (Supplementary Fig. 9c). Interestingly, the most abundant enrichment was observed in genes belonging to metabolic pathways (Supplementary Fig. 9d, Supplementary Data 2). For these four genes, we used the IGV software to determine whether they are predicted targets and, interestingly, the Arlr binding sites were almost all in the CDS regions of the genes (Fig. 5b).

Next, we designed primers based on the highest peaks in the RIP group and performed RIP–qRT-PCR to amplify the bound mRNAs. To prevent indirect binding due to formaldehyde-based cross-linking in the RIP-seq assay, we used a reagent kit without formaldehyde in this assay. The mRNA levels of the four genes immunoprecipitated by anti-GFP were much higher than in the input immunoprecipitated by anti-IgG (Fig. 5c). Meanwhile, the negative control *PGRP* and the positive control *iab-7* showed similar levels between the IgG and RIP group. Altogether, these results suggest that Arlr binds to mRNAs encoding lipolytic proteins and negatively regulates their expression levels.

### DendoU is a functional paralog of Arlr

Examination of the *Drosophila* genome reveals the existence of an Arlr paralog encoded by *dendoU* (*Drosophila endoribonuclease U-specific*). Both genes have the highly conserved EndoU-like domain but differ at the N terminus as DendoU does not contain a signal peptide and the proline-rich and glycine-rich domains (Supplementary Fig. 3a)[50]. Since

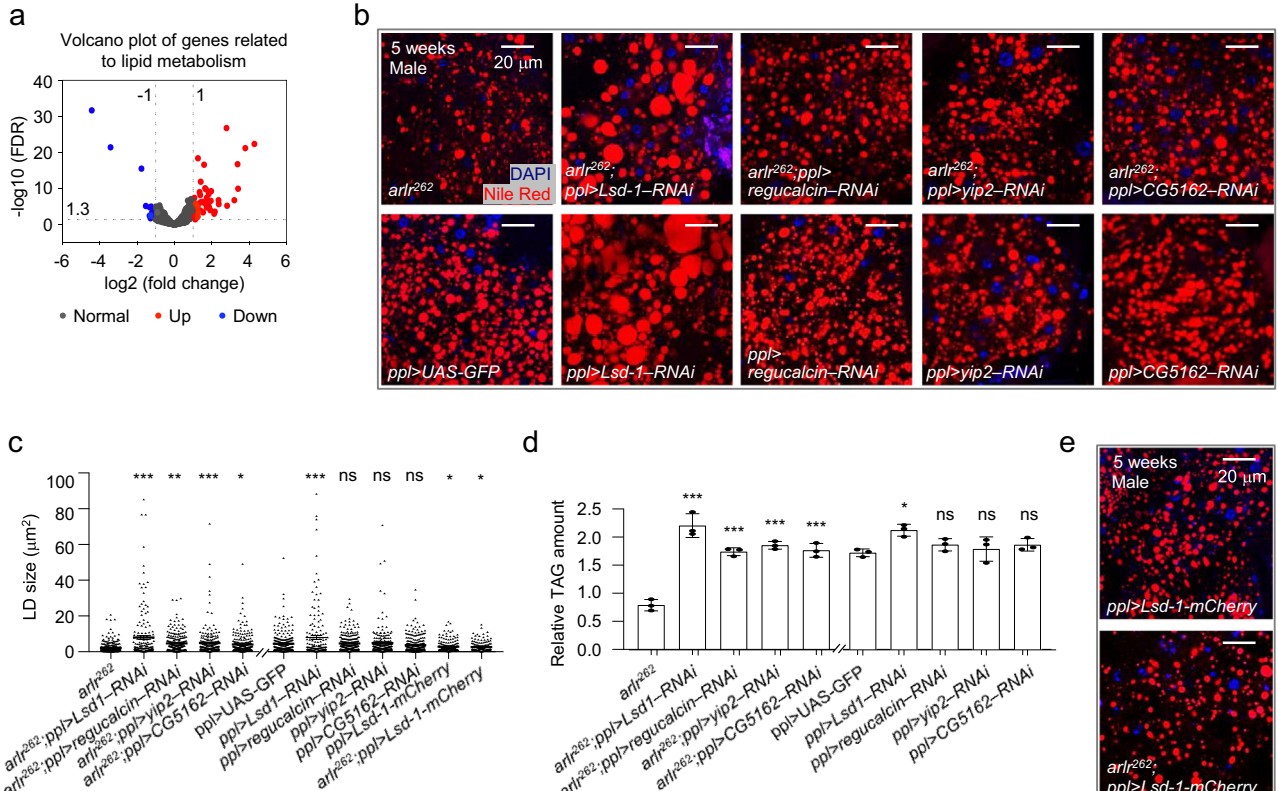

**Fig. 4 | Genetic interactions of *arlr* and lipolytic genes. a** Volcano plot of the lipid metabolism-associated genes. **b** Genetic interactions of *arlr²⁶²* and lipolytic genes in males at 5 weeks. **c** Quantification of LD size. *n* = 6 biologically independent animals. ***P < 0.001 for *arlr²⁶²* vs. *arlr²⁶²;ppl>Lsd1–RNAi*, for *arlr²⁶²* vs. *arlr²⁶²;ppl>yip2–RNAi* and for *ppl > UAS-GFP* vs. *ppl>Lsd1–RNAi*; **P = 0.001; *P = 0.02 for *arlr²⁶²* vs. *arlr²⁶²;ppl > CG5162–RNAi*, *P = 0.05 for *ppl > UAS-GFP* vs. *ppl>Lsd-1-mCherry*, *P = 0.05 for *ppl > UAS-GFP* vs. *arlr²⁶²;ppl>Lsd-1-mCherry*; ns *P > 0.99 for *ppl > UAS-GFP* vs. *ppl>regucalcin–RNAi*, P = 0.99 for *ppl > UAS-GFP* vs. *ppl>yip2–RNAi*, and P = 0.63 for *ppl > UAS-GFP* vs. *ppl > CG5162–RNAi*. Data were

analyzed by Mixed-effects analysis Dunnett's multiple comparison test (two-tailed). **d** Relative TAG amounts. *n* = 3 biologically independent experiments. ***P = 0.001 for *arlr²⁶²;ppl>regucalcin–RNAi* and ***P < 0.001 for other three genotypes; *P = 0.03; ns P = 0.03 for *ppl>regucalcin–RNAi*, P = 0.97 for *ppl>yip2–RNAi*, and P = 0.69 for *ppl>Lsd-1-mCherry*. Data were analyzed by one-way ANOVA with Tukey's multiple comparison test. **e** Adults expressing *Lsd-1* showed subtle LD defects. The statistic of LD size is in (**c**). Panels (**b**–**e**) are from males. Error bars represent SEM. Scale bars are 20 μm.

---

truncated Arlr without the N-terminal signal peptide could restore the LD defects seen in *arlr* mutants, we tested whether *dendoU* could compensate for loss of *arlr* function. Interestingly, *dendoU* expression restored the LD defects in *arlr* mutants at 5 weeks (Fig. 6d, h, j) indicating that DendoU is a functional paralog of Arlr.

As *dendoU* could rescue *arlr* mutants, we examined the phenotypes associated with either loss or overexpression of *dendoU*. In the fat body of third instar larvae with knockdown of *dendoU* (*ppl>dendoU–RNAi*), LDs were normal in size and their number was slightly decreased (Supplementary Fig. 10b, c). In adults, the number of LDs was greatly reduced and did not show age-dependent changes (Fig. 6e, i, Supplementary Fig. 10e, h). Further, consistent with the reduced number of LDs, the TAG amount was reduced (Fig. 6j). Finally, ectopic expression of *dendoU* in the fat body (*ppl>dendoU-HA*, Fig. 6c, h, j) resulted in extra-large LDs, similar to those of *ppl>arlr^ΔSP-HA* flies (Fig. 2e, i, j) but not of *ppl>arlr-HA* flies (Fig. 2c, i, j), indicating that overexpression of the non-secreted EndoU proteins in the fat body is sufficient to increase the size of LDs.

To further test whether *arlr* and *dendoU* are functional paralogs, we expressed *arlr* in *dendoU–RNAi* flies. Arlr could rescue the LD defects associated with loss of *dendoU* (Fig. 6f, Supplementary Fig. 10f), suggesting that the two paralogs have an overlapping function. Thus, we tested whether loss of both *arlr* and *dendoU* would exhibit a stronger phenotype than loss of a single gene. Reducing *arlr* in flies with knockdown of *dendoU* did not increase the severity of loss of *dendoU* in young adults (Supplementary Fig. 6g). However, smaller

and fewer LDs was observed in aging flies (Fig. 6g–j), indicating that Arlr and DendoU have additive effects to regulate total lipid levels.

### Human ENDOU can functionally substitute for Arlr

To determine whether the function of human ENDOU in lipid metabolism is conserved, we generated transgenic flies expressing full-length human ENDOU (*UAS-ENDOU-HA*) (Supplementary Fig. 3a) and expressed human ENDOU in the fly fat body. Human ENDOU also contains a signal peptide, followed by the EndoU domain[60]. Strikingly, when expressing *ENDOU* in the *arlr²⁶²* mutant background, LD sizes and TAG levels were restored (Fig. 7d–f), suggesting that the function of human ENDOU in lipid metabolism is evolutionarily conserved. On the other hand, overexpression of *ENDOU* (*ppl > UAS-ENDOU-HA*) had no effects on LDs, including LD size and TAG levels (Fig. 7c, e, f).

Then, we examined whether ENDOU expression can revert the upregulation of the four proposed target genes in *arlr* mutants. Expressing human *ENDOU* showed decreased expression of the four genes. In the *arlr* mutant background, *Lsd-1*, *CG5162*, *yip2*, and *regucalcin* were all significantly reduced by expressing human *ENDOU* (*CG5162* and *yip2* as examples in Fig. 7g), suggesting that the EndoU family proteins are conserved in regulating the target RNAs.

### Discussion

In this study, we report that high expression of the endonuclease Arlr during aging is essential for lipid accumulation and that in the absence

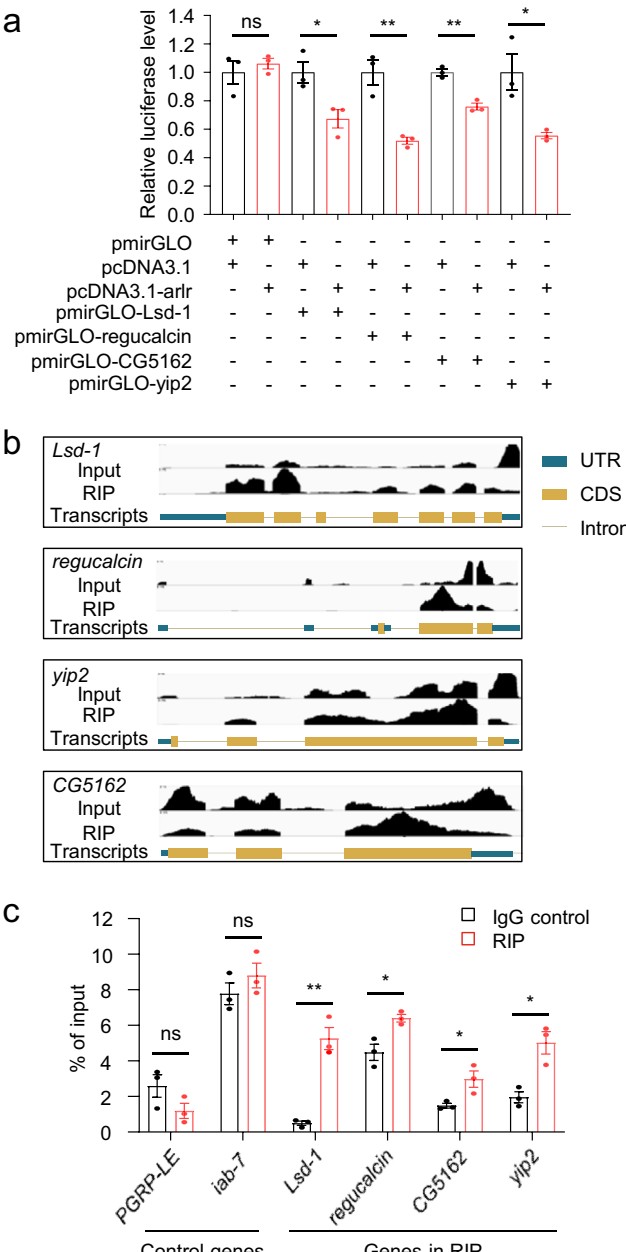

**Fig. 5 | Arlr binds and regulates the mRNA level of lipolysis-associated genes.** **a** Luciferase reporter assay. In the presence of Arlr, the target genes were down-regulated compared to the blank control. $n = 3$ biologically independent experiments. ns $P = 0.54$; *$P = 0.03$; **$P = 0.006$ for pmirGLO-regucalcin and **$P = 0.002$ for pmirGLO-CG5162. **b** Genomic loci analysis of the peaks using the IGV software. Black peaks show the binding ability of Arlr-GFP with selected mRNAs. **c** qRT-PCR results revealed higher relative enrichment of selected mRNAs in Arlr-GFP-immunoprecipitated RNA samples. *PGRP-LE* was a negative control and *iab-7* was a positive control. $n = 3$ biologically independent experiments. ns $P = 0.14$ for *PGRP-LE* and $P = 0.33$ for *iab-7*; **$P = 0.002$; *$P = 0.02$ for *regucalcin*, *$P = 0.04$ for *CG5162*, and *$P = 0.01$ for *yip2*. Data in (**a**) and (**c**) were analyzed by independent two-sample *t* tests (two-tailed). Error bars represent SEM.

of Arlr activity lipid storage in LDs in adult adipose tissues is affected due to an increase in lipolysis. We demonstrate that Arlr binds to the mRNAs of a number of lipolytic genes and negatively regulates their expression and that the endonuclease domain of Arlr is necessary to rescue *arlr* mutants. Altogether, we propose that the endoribonuclease Arlr is required to maintain lipid homeostasis by downregulating lipolytic genes.

Previous studies have implicated the EndoU protein family in lipid metabolism by affecting mRNA levels. The nematode homolog ENDU-2 protects germline immortality via downregulation of genes involved in lipid metabolism. However, in contrast to our findings, the lipid content is increased in *ENDU-2* mutants[61]. As the specific targets of ENDU-2 have not yet been characterized, further studies will be needed to reconcile these observations. In addition, overexpression of the zebrafish or human ENDOU increases human CHOP mRNA translation via cleavage of the upstream open reading frames (uORFs), which negatively affects translation of C/EBP homologous protein (CHOP)[62]. As a transcription factor, CHOP is upregulated by lipid accumulation-induced ER stress and interacts with FoxO to promote hepatic lipogenesis through activation of peroxisome proliferator-activated receptor γ (PPARγ) expression[63]. In addition, lipids are reduced in CHOP knockdown cell lines[64]. However, severe impairment of ER activates CHOP to induce lipoapoptosis[65,66], thus inhibiting adipogenesis, probably through specific target genes[67]. Despite these studies, the role of EndoU proteins in lipid metabolism has remained largely unknown.

EndoU proteins, which contain RNA binding domains, cleave single-stranded RNA harboring U-rich sequences[47–52]. We found that a number of lipolytic genes are negatively regulated by Arlr via its binding to target mRNAs and that the EndoU domain is required for Arlr activity. Strikingly, knockdown of four lipolytic genes, *Lsd-1*, *CG5162*, *yip2*, and *regucalcin*, rescues the LD defects in *arlr* mutants (Fig. 4). Lsd-1 proteins form a scaffold on the surface of LDs and recruit lipases such as HSL to stimulate lipolysis[24]. *Lsd-1* is exclusively expressed in fat bodies and *Lsd-1* mutants display giant LDs both in larvae and adult flies and exhibit adult-onset obesity[26,27]. In contrast, overexpression of *Lsd-1* in the fat body results in small LDs (Fig. 4)[27]. Interestingly, AKH/Glucagon signaling phosphorylates Lsd-1 to stimulate lipolysis in adult flies[26,68,69]. As the AKH receptor is normally expressed in *arlr* mutants (Supplementary Data 1), it will be interesting to test whether Arlr antagonizes AKH. In addition, as cytoplasmic lipase-driven lipolysis acts on large LDs[22], only large-sized medial LDs (mLDs) are regulated by Arlr (Fig. 1), but not small peripheral LDs (pLDs) (Supplementary Fig. 5). This is consistent with the mechanism that mLDs rely on Lsd-1, whereas pLDs are regulated by Lsd-2 and the LD-PM (plasma membrane) contact protein Snazarus[70]. Among the other targets, *CG5162* mRNA levels have been previously reported to increase in response to acute exercise which leads to a significant reduction in LD size[58]. This is consistent with its role in lipolysis. Interestingly, two other major lipases, HSL and Bmm, are not transcriptionally regulated by Arlr (Supplementary Data 1). Further characterization of the epistatic relationships between Bmm, HSL and Arlr will be of interest. Another target, *Yip2*, is a fasting-inducible gene[71] that acts in the catalysis of fatty acid oxidation[57]. Finally, Regucalcin/SMP30 transcriptionally represses the adipokines *leptin* and *adiponectin*[72], but how Regucalcin/SMP30 regulates lipid metabolism is not clear. Altogether, we speculate that EndoU family proteins influence the synthesis of lipolytic proteins by releasing mRNAs from the ER membrane. When Arlr is lost, the function of mRNA degradation is impaired, resulting in accelerated lipolysis of large medial LDs, ultimately leading to fast lipid consumption in aging flies (Fig. 7h).

We find that the two fly paralogs of EndoU, Arlr and DendoU, share similar functions during lipid metabolism. Mutations of either gene lead to reduced fat content. However, while the number of LDs in *dendoU*-knockdown flies is reduced, small LD size is the major defect in *arlr* mutants in aging flies (Figs. 1, 4). In vitro RNA processing assays indicate that DendoU preferentially cleaves at oligo(U), whereas Arlr preferentially cleaves between A and C nucleotides and has less preference for oligo(U)[50]. Moreover, although some residues that are essential for RNA binding and RNA cleavage are conserved in both Arlr and DendoU, others located at the N terminal variable region that are essential for RNA binding are less conserved[50,52,73], suggesting that

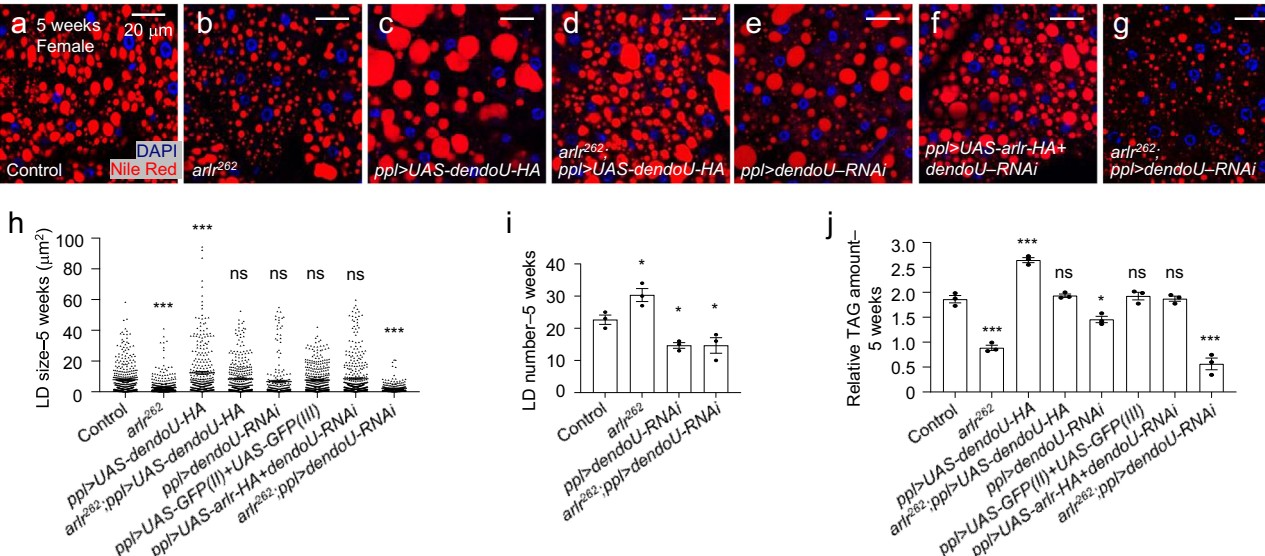

**Fig. 6 | DendoU and Arlr have overlapping function.** All panels in this Figure are from females at 5 weeks. **a, b** LDs in control and *arlr²⁶²* mutant flies. **c** Ectopic expression of *dendoU* in the fat body showed larger LDs. **d** Expression of *dendoU* restored LD defects in *arlr* mutants. **e** Knockdown of *dendoU* showed fewer LDs. **f** Expression of *arlr* restored the LD defects in *dendoU*-knockdown flies. **g** Reduction of both *arlr* and *dendoU* showed more severe LD defects than either mutants. Note that the experiment with *dendoU* RNAi was performed with a single RNAi line so the conclusion may need to be confirmed with additional lines in the future. **h** Quantification of the LD size in the above flies. *n* = 6 biologically independent animals. ****P* < 0.001; ns *P* = 0.83 for *ppl>dendoU-RNAi* and *P* > 0.99 for other three genotypes. **i** Quantification of the LD number per sample. *n* = 3 biologically independent animals. **P* = 0.04, 0.03, and 0.03, respectively. **j** Relative TAG amounts. *n* = 3 biologically independent experiments. ****P* < 0.001; ns *P* > 0.99; **P* = 0.01. Data in (**i**) were analyzed by Mixed-effects analysis Dunnett's multiple comparison test (two-tailed) and others by one-way ANOVA with Tukey's multiple comparison test. Error bars represent SEM. Scale bars in (**a–g**) are 20 μm.

DendoU and Arlr have both common and distinct molecular targets. Despite these differences, expression of either gene is sufficient to rescue the LD defects in each mutant (Figs. 2, 6). Further, a short period of starvation induces moderate expression of *arlr* and down-regulation of *dendoU* in fat bodies[74], indicating a compensatory regulation of these two EndoU genes to keep the balance of lipid metabolism. In addition, like *dendoU*, expressing human *ENDOU* can compensate for loss of *arlr* (Fig. 7). Taken together, our findings reveal a conserved role of EndoU family proteins in lipid metabolism and that these proteins may function through distinct targets but have complementary roles in lipid accumulation.

Interestingly, EndoU proteins in most species contain a signal peptide at the N terminus (Supplementary Fig. 2)[60] and can be secreted[55,61]. Previously, we reported that knockdown of *arlr* in the fat body was associated with poor climbing ability consistent with defects in muscle activity and Ubiquitin protein aggregates in muscles[55]. In addition, Arlr could be detected at the surface of thoracic muscles suggesting that fat body derived Arlr directly affect muscle activity. Further, in *C. elegans*, secreted EndoU from the soma has been found to protect germline immortality[61]. Intriguingly, overexpressing *dendoU*, which has no signal peptide sequence, leads to enlarged LDs, similar to *arlr*^ASP (signal peptide-deprived Arlr). However, over-expressing full-length *arlr* or human *ENDOU* show normal LDs (Figs. 2, 7), which is likely due to its secretory capacity. Further studies will be required to characterize the exact role of secreted EndoU proteins.

In aging flies, the highest DEGs are in the fat body and several adipose cell types, highlighting that major changes occur in fat cells during aging[54]. *arlr* is highly expressed during adult stages and snRNAseq analysis of the FCA and AFCA data sets shows that the expression of Arlr is increased in aging fat cells (Fig. 1, Supplementary Fig. 1). Consistent with the high levels of Arlr, expression of the target senescence marker gene *regucalcin/SMP30* decreases with age in rats[75], indicating that Arlr could be a marker of aging. Arlr attenuates lipolytic gene expression, resulting in lipid homeostasis in aging flies (Figs. 4, 5). Aging-associated reduction of lipolysis is mediated by inhibition of lipolytic pathways, such as the decrease of HSL and ATGL with age[33,34]. Further, promoting lipolysis by genetic manipulations and dietary restriction (DR) is considered to be associated with lifespan extension[76]. Upon DR, flies shift their metabolism toward increasing fatty-acid synthesis and breakdown, and disruption of lipid synthesis or oxidation inhibits lifespan extension upon DR, indicating that lipid homeostasis is essential for lifespan extension[42]. In *arlr* mutants, lipid consumption is severely accelerated, indicating a disruption of lipid homeostasis, thus lifespan is reduced under standard food and starvation conditions (Figs. 1, 3). Excessive nutrition replenishes the lipid content and rescues the short lifespan of *arlr* mutants (Fig. 3). Thus, Arlr is essential for longevity by promoting the balance of lipid metabolism via lipolytic regulation.

EndoU family of endonucleases have been implicated in many processes, including as a tumor biomarker in human beings[77–79], immune response in mice[51], ER morphology in *Xenopus*[80], neurodegeneration in *Drosophila*[50], cold tolerance, nucleotide metabolism, lifespan and germline immortality in *C. elegans*[61,81,82], and viral replication and pathogenicity[83–85]. Our findings reveal another role of EndoU family proteins in lipolysis and provide mechanistic insights for such a specific function. Understanding of EndoU in lipid metabolism may provide insights for the treatment of lipid metabolism-associated diseases and promotes healthy aging.

## Methods

### *Drosophila* stocks and rearing conditions

Flies were reared at 25 °C with a 12:12 h light/dark cycle on standard food unless otherwise mentioned. Ingredients of various foods are included in Supplementary Table 1. For the starvation, high sugar, and high-fat diet assays, flies were raised on standard food and then transferred to the relevant diet for 24 h before dissection of the fat body and TAG measurements. For lifespan tests, flies were raised on standard, poor nutrition, or high nutrition food throughout the adult stage. *P{nos-Cas9.R}attP2* is the background stock (control) of deletion mutants (Bloomington *Drosophila* Stock Center/BDSC 78782). For

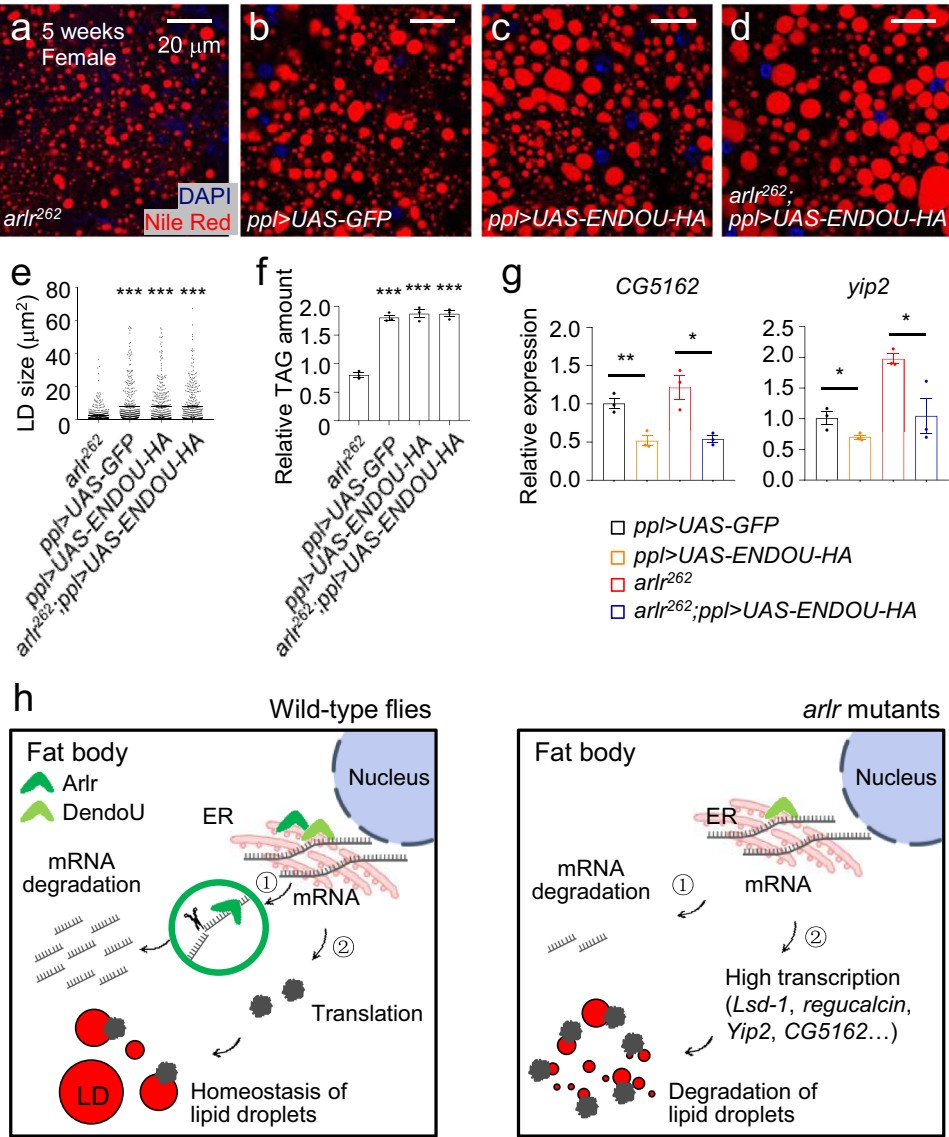

**Fig. 7 | Human ENDOU rescue *arlr* mutants.** Panels **a**–**e** are from females at 5 weeks. **a**, **b** LDs in *arlr^262* mutant flies and the *ppl > UAS-GFP* control. **c** Ectopic expression of human *ENDOU* in the fat body showed normal LDs. **d** Expression of human *ENDOU* restored the LD defects in *arlr* mutants. **e** Quantification of LD size. *n* = 6 biologically independent animals. ***P* < 0.001. **f** Relative TAG amounts were restored by human *ENDOU*. *n* = 3 biologically independent experiments. ****P* < 0.001. **g** *CG5162* and *yip2* were reduced by expressing human *ENDOU*. *n* = 3

biologically independent experiments. ***P* = 0.007; ***P* = 0.01, 0.05, and 0.03, respectively. Data were analyzed by one-way ANOVA with Tukey's multiple comparison test in (**d**) and (**e**), and by independent two-sample *t* tests (two-tailed) in (**g**). Error bars represent SEM. Scale bars in (**a**–**c**) are 20 μm. **h** Model: In wild-type flies, EndoU binds to the mRNAs of lipolysis genes and degrades them (pathway 1) to keep the homeostasis of LDs (pathway 2). In *arlr* mutant flies, these mRNAs fail to be degraded, leading to accelerated lipid consumption.

targeted knockdown, overexpression and rescue experiments, the fat body-specific Gal4 drivers *ppl-Gal4* (BDSC 58768) and *LPP-Gal4*[55] were used. *UAS-GFP* (BDSC 4775, 4776) driven by *ppl-Gal4* was used as control. *arlr−RNAi^61997* (BDSC 61997) and *arlr−RNAi^14874* (Vienna *Drosophila* Resource Center/VDRC v14874) were used to knockdown *arlr*. *dendoU−RNAi* (VDRC v9916) was used to knockdown *dendoU*. *Lsd-1−RNAi*, *regucalcin−RNAi*, *yip2−RNAi*, and *CG5162−RNAi* (TsingHua Fly Center/THFC TH04820.N, THU1799, THU3344, and TH03231.N, respectively) were used for the genetic interaction tests with *arlr* mutants. *UAS-KDEL-mCherry*, used to label the ER, was a gift from Yixian Cui (Wuhan University, Wuhan, China).

**CRISPR/Cas9 mediated generation of *arlr* mutants and transgenic strains**
The *arlr* deletion mutants and UAS transgenic flies were established at the *Drosophila* resources and technology platform, Core Technology

Facility of Center for Excellence in Molecular Cell Science, Chinese Academy of Sciences. sgRNAs targeting *arlr* for CRISPR/Cas9-midiated targeted mutagenesis were designed with the tools available at https://zlab.bio/guide-design-resources. Two sgRNAs (5'-ggacgcagtgatggc-cacgc-3' and 5'-gccagtgaatactgtgtttg-3') were injected into *P{nos-Cas9.R} attP2* embryos. Putative transformants were crossed to the balancer strain (*yw; Sp/CyO; MKRS/TM2*) for 7 days. Then, DNA was extracted for PCR detection. Primers flanking the predicted lesions were designed for PCR. Two individual mutants were identified, *arlr^262* (deletion of 1261–1523 bp in the coding sequence, homozygous viable) and *arlr^364* (deletion of 1255–1619 bp in the coding sequence, homozygous viable).

UAS transgenic lines, including *UAS-arlr-HA* (encoding the full length 592 a.a.), *UAS-arlr^ΔSP-HA* (encoding a truncated protein lack of the N-terminal 75 a.a. including the signal peptide), and *UAS-arlr^N-HA* (encoding a truncated protein of 1−392 a.a.) were cloned by PCR. Primers are listed in Supplementary Table 2. For *UAS-ENDOU-HA* and *UAS-*

*dendoU-HA*, the coding sequences of human *ENDOU* and *Drosophila dendoU* were de novo synthesized by Tsingke Biotechnology (Beijing, China). Sequences were inserted into the pUASTattB vector through XbaI and EcoRI sites. All plasmids were injected into *M{vas-int.Dm}ZH-2A w\*; P{CaryP}attP2* (BDSC #68A4) embryos. Potential transformants (red eye) were crossed with the balancer strain (*yw; Sp/CyO; MKRS/TM2*) for 7 days. Subsequently, red-eye flies were picked out for DNA extraction and PCR detection.

The *arlr-GFP* transgenic strain was established by Qidong Fungene Biotechnology (Jiangsu, China). sgRNAs (5'-cctcattggcagcgcctatccgg-3' and 5'-agcgcctatccggagatttgagg-3') were designed by Chopchop (http://chopchop.cbu.uib.no/). *Cas9* mRNA was transcribed using the template of a linearized plasmid containing the *Cas9* cDNA (Addgene plasmid 42251). Homology 5' and 3' arms were amplified from the genomic DNA of *arlr* and linked to the backbone of the donor vector pBluescirpt SK vector (pBS) which we named "pBS-Arlr-arm". pBS-Arlr-arm was linearized by PCR and linked to the GFP-loxp-3p3-RFP-loxp cassette (loxp-3p3-RFP-loxp was a marker with red eyes) to form the final donor construct "pBS-Arlr-GFP-LRL". A formula of 30 μL containing 7.5 μg sgRNA, 15 μg Cas9 mRNA, and 9 μg donor plasmid in DEPC water was injected into *w^{1118}* embryos. P0 progenies were crossed with flies carrying the *FM7a* balancer. F1 flies were screened for RFP expression in the eyes. PCR was performed to validate the F1 flies. F2 flies from F1-positive tubes were crossed with Cre stocks (*yw,cre*) to remove the RFP marker between the two loxP sites. Flies without RFP were balanced with *FM7a* and kept for further study. All primers for PCR detection are listed in Supplementary Table 2.

## Immunostaining and confocal microscopy

Fat bodies were dissected in ice-cold PBS and fixed in 4% paraformaldehyde at room temperature with gentle shaking for 15 min and washed in PBS for 15 min. Subsequently, the tissues were incubated with BODIPY493/503 (1:1000, D3922, Invitrogen, Eugene, USA) or Nile Red (1:1000, 72485, Sigma, St. Louis, USA) for 30 min at room temperature with gentle shaking in the dark and then washed in PBS for 30 min. Samples were mounted in Vectashield containing DAPI (Vector Laboratories, Burlingame, USA). For rabbit anti-GFP (A11122, 1:2000, Invitrogen) and HA-Tag (C29F4) rabbit mAb (1:8000, #3724, Cell Signaling, Danvers, USA) staining, tissues were incubated with the primary antibody overnight at 4 °C, washed in PBS for 30 min, followed by the secondary antibody goat anti-rabbit Alexa 647 (A21245, 1:200, ThermoFisher, Eugene, USA) or goat anti-rabbit Alexa 488 (A21206, 1:200, ThermoFisher) for 1 h, washed in PBS for 30 min and finally mounted in Vectashield containing DAPI. Confocal images were acquired using Zeiss LSM 800 confocal microscope with a 63X oil objective. Sequential scanning was performed every 0.2 μm for the reconstruction of the x-z and y-z sections. For immunostaining and the following experiments, including dietary tests, triglyceride, glucose and trehalose measurements, and lifespan, virgin females and males were measured separately.

## Triglyceride, glucose, and trehalose measurements

For triglyceride measurements, ten adults were homogenized in 500 μL PBS containing 0.2% Triton X, heated at 70 °C for 5 min, and centrifuged at 18,400 × g for 10 min. Ten microliters of supernatant was used to measure the TAG level by Serum Triglyceride Determination Kits (TR0100-1KT, Sigma). Protein amounts were measured using Bradford Reagent (P0006, Beyotime, Shanghai, China). TAG level was normalized to the protein amount.

For glucose measurements, ten adult flies were homogenized in 200 μL distilled water after being weighed and centrifuged at room temperature at 8000 × g for 10 min. The supernatant was used to measure the glucose level using Glucose Detection Kits (BC2505, Solarbio, Beijing, China). For trehalose measurements, ten adult flies were weighed and homogenized in 200 μL extracting solution. After

standing at room temperature for 45 min, extractions were centrifuged at room temperature at 8000 × g for 10 min. The supernatant was used to measure the trehalose level using Trehalose Detection Kits (BC0335, Solarbio). The glucose and trehalose levels were normalized to the weight. Each genotype was measured three times.

## Lifespan measurements

After eclosion, flies were reared in 4–8 vials (30 adults in each vial) at 25 °C and transferred into new vials every three days. The number of survival individuals was recorded at each transfer. The survival rate was calculated by the percentage of total surviving flies. Surviving curves were generated using GraphPad Prism 9.5. Lifespan test was repeatable for each genotype and one of the two repeats was shown in the figures.

## Transcriptome sequencing (RNA-seq)

Total RNA of control and *arlr^{262}* mutant flies including equal number of both males and females (20 adults in total) were extracted from the abdomen of 5-week adults discarding the intestines and ovaries using TRIzol™ Reagent (15596026, Invitrogen). Transcript libraries were constructed via Illumina novaseq 6000 (Illumina). Each treatment had 3 biological replicates. DESeq was used to analyze the differentially expressed genes (DEGs). Fold change ≥2 and false discover rate (FDR) < 0.01 were considered as significantly changed expression.

## RNA immunoprecipitation sequencing (RIP-seq)

Abdomens from 20 *arlr-GFP* 5-week-old adult flies including equal number of males and females were dissected in ice-cold PBS, intestines and ovaries were discarded, and samples were collected in 1 mL ice-cold PBS. Tissues were centrifuged at 2000 × g for 2 min at 4 °C. Sediments containing RNAs and Arlr-GFP proteins were cross-linked by 180 μL 37% formaldehyde at room temperature and gently shaken for 15 min. To stop cross-linking, 20 μL glycine with a final concentration of 1.25 M was mixed using a vortex for 5 min. After centrifugation at 2000 × g for 2 min at 4 °C, the supernatant was discarded. Sediments were washed in 1 mL buffer A1 (60 mM KCl, 15 mM NaCl, 4 mM MgCl$_2$, 15 mM HEPES, 0.5% Triton X-100, 0.5 mM DTT, 10 mM sodium butyrate, 1:100 protease inhibitor, 100 U/mL RNase inhibitor) before centrifugation at 2000 × g for 2 min at 4 °C and 1 mL lysis buffer 1 (140 mM NaCl, 15 mM HEPES, 1 mM EDTA, 0.5 mM EGTA, 1% Triton X-100, 0.5 mM DTT, 0.1% sodium deoxycholate, 10 mM sodium butyrate, 1:100 protease inhibitor, 100 U/mL RNase inhibitor) before centrifugation at 2000 × g for 2 min at 4 °C. Tissues were resuspended in 300 μL 1% SDS lysis buffer 2 (0.5% N-lauroylasarcosine in fresh lysis buffer 1) and placed on ice for 1 h. Lysates were sonicated for 7.5 min on ice (using a setting of 30 s on–60 s off, 5 cycles, moderate intensity) to shear the chromatin into -0.5 kb fragments. Sonicated lysates were centrifuged at 12,000 × g for 2 min at 4 °C, then the supernatant was retained in a new enzyme-free tubes.

Fifty μL of the above solution was stored at −80 °C as an input control for later use. The remaining 250 μL solution was added with 2 μg anti-GFP antibody (A11122, Invitrogen) as the IP group, remaining at 4 °C for 4 h with gentle shake. The solution was pre-washed using 30 μL protein A beads (10001D, Invitrogen) at 4 °C overnight, and then washed by lysis buffer wash (0.05% SDS in fresh lysis buffer 1, 1:100 protease inhibitor, 100 U/mL RNase inhibitor) for 4 times, 5 min each, followed by TE (0.1 mM EDTA, 10 mM Tris HCl pH = 8.0, 1:100 protease inhibitor, 100 U/mL RNase inhibitor) for 2 times. The IP sample was incubated in 100 μL elution buffer 1 (10 mM EDTA, 50 mM Tris-HCl pH = 8.0, 1% SDS, 1:100 protease inhibitor, 100 U/mL RNase inhibitor) at 130 rpm for 15 min at 65 °C. The incubation of supernatant was then repeated in 150 μL elution buffer 2 (0.1 mM EDTA, 10 mM Tris-HCl pH = 8.0, 0.67% SDS, 1:100 protease inhibitor, 100 U/mL RNase inhibitor). Next, both input and IP group samples were treated with 20 μg protease K at 130 rpm for 4 h at 65 °C for the reversal of cross-linking. Solutions with 1/10 volume of 3 M NaAC (with 0.3 M final

concentration, pH = 5.2), 2.5 volume of pre-cold ethanol, 1 μL glycogen and 3 μL RNase inhibitor were used at −80 °C overnight for RNA purification before centrifugation at 12,000 × $g$ for 30 min at 4 °C. The precipitate was washed with pre-cold 75% ethanol twice before being dissolved in 20 μL enzyme-free water. The Ribo-off rRNA Depletion Kit (N406-01/02, Vazyme, Nanjing, China) was used before RIP-seq to exclude the interference of ribosomal RNA. RIP-seq was performed by Novogene (Beijing, China) using an Illumina Novaseq 6000 platform. There were 2 biological replicates for the input and IP groups. For genomic loci analysis, the bam raw document was analyzed using the Integrative Genomics Viewer (IGV) software (http://software. broadinstitute.org/software/igv/, California, USA). Potential motifs in the target sequences were predicted using Homer software (V4.11, University of California at San Diego, http://homer.ucsd.edu/homer/).

### Quantitative real-time PCR (qRT-PCR)
Following extraction of total RNAs, cDNA was synthesized using PrimeScriptTM Reagent Kit (RR047A, TaKaRa, Beijing, China) and qRT-PCR was performed using PerfectStartTM Green qRT-PCR SuperMix (+Dye II) (TransGen, Beijing, China, #AQ132-24) using the QuantStudio 6 Flex platform. To prevent cross-linking and verify that the target genes directly bind to Arlr, we performed RIP–qRT-PCR analysis using a reagent kit without formaldehyde (RNA Immunoprecipitation [RIP] Kit, Catalog Bes5101, BersinBio, Guangzhou, China). These qRT-PCR data were normalized to the *actin* control. Other data were normalized to the internal control *Rpl32*. Primers are listed in Supplementary Table 2. Each genotype had three biological replicates and three technical replicates.

### Luciferase reporter assay
Full-length cDNAs of *Lsd-1*, *regucalcin*, *yip2*, and *CG5162* were cloned into the pmirGLO dual-luciferase miRNA target expression vector (E1330, Promega, Beijing, China). The full-length cDNA of *arlr* was cloned into a pcDNA3.1 vector (V790-20, Invitrogen). HEK293T cells (CRL-3216, ATCC, USA) were cultured in DMEM medium (11965-092, Gibco, New York, USA) with 10% fetal bovine serum (11011-8611, Biobase, Jinan, China) and 1% antibiotic (15140-122, Gibco), and maintained in incubators with 5% $CO_2$ at 37 °C. Cells were seeded in a 24-well plate. Transfection was performed at 70–90% confluence with Lipo8000 transfection reagent (C0533, Beyotime). Luciferase assays were performed 2 days after transfection using Dual-Lumi II luciferase reporter gene assay kit (RG089S, Beyotime). Results were tested by SpectraMax i3x (TECAN, Switzerland).

### Western blot
Whole adults were homogenized in ice-cold RIPA buffer (P0013B, Beyotime) supplemented with 100 μM protease inhibitors PMSF (ST506, Beyotime). Total protein concentrations were measured using the BCA assay Kit (P0012S, Beyotime). Protein samples were diluted to equal concentrations using lysis buffer and loaded into SDS-PAGE gels for Western blot analysis. The primary antibodies were HA-Tag (C29F4) rabbit mAb (1:8000, #3724, Cell Signaling) and mouse monoclonal anti-α-tubulin (1:50,000, Sigma, T6074). The secondary antibodies were goat anti-rabbit IgG H&L (HRP) (1:20,000, ab205718, Abcam, Cambridge, UK) and goat anti-mouse IgG (H + L), HRP (1:20,000, BE0102, Easybio, Beijing, China).

### Statistical analysis
Images were processed with Photoshop 2021. To quantify LD size, Image J V1.53 was used to measure the area of LDs on the same focal plane around the nuclei. For each genotype, a total of three adipocytes were measured in each fly sample and six flies were used. Lifespan data were subjected to survival analysis using a log-rank Mantel-Cox test and presented as survival curves. Statistical charts were produced by GraphPad Prism 9.5. Significance between two genotypes was determined by two-tailed Student's *t* test, whereas multiple comparisons between genotypes were determined by one-way ANOVA with Tukey's or Mixed-effects analysis Dunnett's multiple comparison test.

### Reporting summary
Further information on research design is available in the Nature Portfolio Reporting Summary linked to this article.

## Data availability
The RNA-seq data generated in this study have been deposited in the Genbank database under accession code PRJNA943130. The ChIP-seq data generated in this study have been deposited in the Genbank database under accession code PRJNA943378. Gene expression images are available at the Aging Fly Cell Atlas platform (https://hongjielilab. shinyapps.io/AFCA/). Source data are provided with this paper.

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

## Acknowledgements
We thank Yixian Cui for the *UAS-KDEL-mCherry* flies, the Bloomington *Drosophila* Stock Center, TsingHua Fly Center, and the Vienna *Drosophila* Resource Center for providing fly stocks. We acknowledge Xun Huang, Suning Liu, Junzheng Zhang, Shuo Yan, Hang Zhou, Stephanie Mohr, Ahmad Muhammad and Yong Q. Zhang for discussions and constructive suggestions. This work was supported by the National Natural Science Foundation of China (32030012 to J.S. and 31872293 to D.W.). Work in N.P. laboratory is supported by the Howard Hughes Medical Institute.

## Author contributions
D.W. conceived the project. X.S., D.W., and X.K. performed the experiments. J.S. supervised the experiments. D.W., X.S., J.S., and X.K. analyzed the data. D.W. wrote the manuscript. N.P. provided comments and edited the manuscript.

## Competing interests
The authors declare no competing interests.
