## [Peer Review File · Nature Communications]

The endoribonuclease Arlr is required to maintain lipid homeostasis by downregulating lipolytic genes during agingReviewers' Comments:

Reviewer #1:

Remarks to the Author:

Sun et al. investigated the function of Arlr, a fly homolog of ENDOU, in lipid metabolism using the *Drosophila* model system. They showed a higher expression of Arlr in the *Drosophila* fat body together with an age-dependent upregulation. The authors clearly revealed that the arlr mutants increased lipolysis without affecting lipogenesis, reduced lipid droplet size, and shortened lifespan, which Arlr RNAi and rescue experiments further confirmed. In addition, they found that a functional paralog of arlr, CG3303, and human ortholog, ENDOU can functionally substitute for Arlr in the regulation of lipid metabolism. Since the physiological role of the ENDOU protein family in controlling lipid homeostasis has not been extensively studied, this work contributes to our understanding of the functional relevance of ENDOU in this context.

In the mechanism study part, the authors found that the EndoU-like domain is essential for Arlr's role in lipid metabolism. They also identified genes differentially expressed in arlr mutants and mRNAs associated with Arlr. Among the lipid metabolism-related genes identified were Lsd1, regucalcin, yip2, and CG5162, whose mRNAs are bound by Arlr and upregulated in arlr mutants. Knockdown of these genes alleviated the reduction in lipid droplet size in arlr mutants. Based on these findings, the authors proposed the following model: Arlr affects lipid metabolism by degrading the mRNAs of lipolysis genes.

The findings on the physiological role of Arlr in lipid metabolism are solid and strongly supported by intensive phenotypic analysis. However, the primary limitation of this study is the absence of clear evidence on the molecular mechanism of Arlr. Further investigation is needed to explore the connection between Arlr's endoribonuclease activity and its role in controlling lipid homeostasis.

Major points:

1. The abstract contains an overstatement in the assertion that "Arlr affects lipid metabolism through the degradation of the mRNAs of lipolysis genes," as there is no direct evidence presented in the manuscript for the molecular mechanism of Arlr. Moreover, the proposed model in Figure 7f is oversimplified, and additional experiments are needed to clarify the precise mechanism involved.
2. The gene expression analysis of wild-type and arlr mutant revealed 580 up-regulated genes and 384 down-regulated genes. Additionally, the RNA Immunoprecipitation experiment identified the binding of Arlr to mRNAs of 1567 genes. However, the authors of the study only examined lipid-metabolism-related genes and ultimately selected four genes as the primary targets to elucidate the molecular mechanism of Arlr. This approach may be susceptible to prejudgment or bias, and a proper in silico analysis should have been conducted to provide the rationale for focusing on specific target genes.
3. Although Arlr was found to be mainly localized in the endoplasmic reticulum (ER) and to regulate ER morphology, these findings were not directly linked to Arlr's regulatory role in lipid metabolism. Given that abnormal ER networks can be observed prior to lipid droplet defects, it is necessary to investigate how Arlr, localized on the ER, regulates lipid droplet size to bridge this gap. As such, the current data cannot conclusively support the proposed model in which Arlr binds to target mRNAs on the ER and degrades them. Further investigations are warranted to elucidate the precise mechanism of Arlr's involvement in lipid metabolism regulation.
4. While the expression levels of four target genes were found to be upregulated in arlr mutants, the direct degradation of target mRNAs by Arlr was not validated in this study. To clarify this point, a reporter assay in an in vitro system could be performed.
5. In the Methods section, the authors stated that they performed RNA immunoprecipitation (IP) sequencing using formaldehyde-based crosslinking IP experiments. While formaldehyde-based crosslinking is a well-known method for investigating protein-bound RNAs, it has unique features that should be taken into account when interpreting the data. For example, formaldehyde-based crosslinking IP efficiently fixes protein-protein interactions, which can result in the indirect binding of

RNAs to a target protein. Therefore, the experimental method used in this study should be clearly specified in the main text, and the results should be presented carefully along with any possible drawbacks of the method.

6. The absence of appropriate controls in Figures 3 and 7a-e makes it difficult to clearly establish the epistatic relationships between different genotypes. Do *ppl>UAS-GFP* or *ppl>UAS-ENDOU-HA* show the same normal phenotype as the wild type or *ppl>+*? The appropriate controls provide a baseline against which the phenotypes of the experimental genotypes can be compared, allowing for a more robust interpretation of the results.

Minor points:

1. Please add p-values to Supplementary Figure 8c.
2. The authors performed a crosslinking-IP analysis to investigate Arlr-bound RNAs, but they used the term "RNA-chromatin immunoprecipitation sequencing". This term is no longer widely accepted and has been replaced by RIP-seq or CLIP-seq.

Reviewer #2:

Remarks to the Author:

The organelle lipid droplet plays key roles in maintaining lipid homeostasis and energy supply. The LD dynamics and metabolism is associated with many disorders, as well as aging or reproduction. Many proteins or signalings balance the LD synthesis and degradation to regulate lipid metabolism. In this manuscript, Sun et al., attempt to explore the essential functions of CG2145, an ortholog of mammalian endoribonuclease that they named *arlr*, on lipid metabolism. They first generated two mutant lines of *arlr* using CRISPR-Cas9. *Arlr* mutant resulted in reduction of LD size and TAG amount but not carbohydrate, which affect lifespan. *Arlr* influences lipid consumption but not lipogenesis. EndoU-like domain is necessary for *Arlr* function in LDs. Another EndoU protein CG3303 and *arlr* are partially redundant in regulating LD size and TAG amount. The *arlr* proteins localized at ER and loss of *arlr* affected ER morphology. The RNA-seq and RIP-seq analysis showed that lipolysis related genes increased due to the loss of *arlr*. *Drosophila* and human ENDOU proteins played conserved role in regulating lipolysis. Although the data presented are intriguing, there are several issues that should be addressed.

Major issues:

1. The FlyAtlas data showed that CG2145 also expressed in many tissues like head. In Laneve's article (reference 49), they also confirm it. So, I wonder whether RNAi of *arlr* in fat body have the similar effect on LD size or lifespan compared to *arlr* mutants. Multiple Gal4 lines should be used, such as LPP-Gal4, *ppl*-Gal4 or *Lsp2*-Gal4. In addition, whether loss of *arlr* in brain cells have no non-autonomous effect on LD size in fat body. Meanwhile, more LD dyes need to be examined in Fig 1b, such as BODIPY or lipidTox.
2. During adult fly development, I ask if *arlr* is regulated by nutritional status, starvation or high nutrition. Starvation speed the lipid consumption in fat body of *arlr* compared with the control. If so, overexpression of *arlr* in fat body would antagonize lipolysis and exhibit larger LD size. Overexpression experiment should be performed.
3. Mutant of *arlr* have no significant effect on larvae or young adult fly development including size and lipid storage, how about the body weight? Or whether starvation also speed lipid consumption in fat body during larval stage?
4. Although the authors have proved that *arlr* and CG3033 are partially functional redundancy using genetic analysis. The mechanism is still unclear. For example, does *arlr* interact with CG3033 to form a complex? Or colocalization?

5. Since the authors found that arlr localized at ER using KDEL-mCherry marker, the figs are not clear. It need more ER markers and electron microscope to confirm it. Since ER is the important organelle of lipid de novo synthesis, why ER morphology change by arlr mutant have litte effect on lipogenesis. Is it contradictory?

6. Similar to lipolysis, lipophagy also play important role in lipid degradation. So, whether loss of arlr also promote lipophagy?

Reviewer #3:

Remarks to the Author:

Sun and colleagues address lipid metabolism regulation in aging using the *Drosophila* model. In this context they functionally characterize the endoribonuclease Age-related lipid regulator (Arlr) (and to some extent the paralog EndoU/CG3303). The study convincingly demonstrates the function of arlr in lipid storage, its implications on lifespan and the role of the gene in shaping the ER. The authors use a combination of RNA-seq and RIP-seq to screen for possible Arlr targets and test selected candidate genes involved in lipid catabolism for their capacity to revert the arlr mutant phenotype. Finally, they provide evidence that the human ENDOU ortholog reverts the arlr mutant phenotype in support of an evolutionarily conserved function of these genes in lipid metabolism.

This highly interesting study presents a comprehensive and largely solid dataset. Yet, as outlined below, some experimental additions are required to convincingly prove the conclusions of the authors in particular with respect to the identified target genes.

Major points:

1) The authors argue that lipogenesis is not affected by the absence of arlr, which is surprising given the severe ER phenotype of the mutants. Their conclusion is based on mutant phenotype reversion in response to short-term feeding with HSD and HFD.

However, incorporation of labelled substrates into TAG is required to make this statement solid. In addition, since LDs are produced at the ER, the localization of lipogenesis proteins will be informative. The observed reversion of the LD and TAG content phenotypes might be due to expansion of pre-existing LDs, which reflect only part of the lipogenesis capacity of the fat body. Essentially, we need to learn to what extent the observed effects on lipid metabolism are indirect effects of ER remodeling in the absence of Arlr.

2) For the EndoU/CG3303 experiments important information and controls are missing. Apparently just one RNAi line was used of unknown knockdown efficiency. Is EndoU/CG3303 expressed in the adult fat body? In the reversion of the EndoU/CG3303 RNAi experiment by co-expression of arlr-HA two UAS transgenes are driven by the ppl Gal4 transgene but a specific control for this scenario is missing. Moreover, Arlr and EndoU/CG3303 have different endonucleolytic specificities (in vitro). How does go together with the observed rescue of arlr by EndoU/CG3303 (over)expression?

3) The statement in line 255 ff. is certainly not true given that mdy, which encodes the major TAG synthetic enzyme is among the DEG (upregulated; although not ≥ 2) and also identified in RIP.

4) The logic concerning Lsd-1 as one of the endonucleolytic targets of Arlr is not conclusive. It is unclear if Arlr can cleave Lsd-1 mRNA in vitro (which applies to all predicted direct targets). Nor it is clear if the increased Lsd-1 mRNA levels in arlr mutants translates to increased protein levels. And if so, whether increased LSD-1 would cause the small/reduced LD arlr mutant phenotype in adult fat body (Bi et al., report on larval fat body only). Methods and tools are published to address these question. The Lsd-1 data shown i.e. the increased LD size upon Lsd-1 RNAi in control or arlr mutants are fully compatible with an arlr-unrelated function of Lsd-1 in LD biology. Expression of arlr deltaSP

increases LD size and TAG content. Does it reduce Lsd-1 mRNA and can these phenotypes be reverted by Lsd-1 co-overexpression?

5) The evidence for the remaining three predicted direct Arlr targets is even weaker, as LD size distributions are relatively subtle (though statistically significant; Fig. 6) and all individual driver and effector controls in the wt and the arlr background are missing. According to Fig. S4i relative TAG amount differences in males at week 5 are very pronounced between controls and arlr mutants. Adding data using this method for all genotypes shown in Fig. 6c would make the conclusions much stronger.

6) Line 307: Fig. 7a-e lacks the control (corresponding to Fig. 1) to appreciate if expression of human ENDOU really does not change LD size and body TAG content over control levels (like arlr-HA but unlike arlr delta SP-HA) as claimed by the authors.

Moreover, if human ENDOU function in arlr mutants reflects evolutionary relationship, the same or at least a substantially overlapping set of target genes are expected to be regulated. Does ENDOU expression revert the up-regulation of the four proposed target genes in arlr mutants?

7) Lifespan experiments:

a) How often were they independently reproduced? Shown is one experiment each.

b) Statistics is missing (median lifespan, statistical test).

c) How is it possible that survival rates increase (see Fig. 1h and 2e shortly after day 40)?

d) Survival curves of controls on standard food (Fig. 1h) and high nutrition diet (Fig. 2e) look remarkably similar; but more rich food appears to reduce median lifespan in contrast to the mutant. How can this be explained?

e) Dietary restriction extends lifespan. Why do controls live shorter under poor nutrition diet? The discussed "unbalanced lipid homeostasis" deserves more explanation.

Minor points:

8) Line 20 and elsewhere: arlr not Arlr (both italics) for the gene abbreviation.

9) Line 48: DGAT1 is omitted as central lipogenesis gene, despite the fact that the Drosophila ortholog is identified in this study as potential Arlr target.

10) Line 66: "TAG synthesis enzyme perilipin"?

11) Line 71: Given that adult Drosophila HSL mutants do not show a neutral lipid phenotype this statement is questionable.

12) Line 128ff: What does the LD size distribution and number in 1 week old controls to 3 week old controls look like? How does this relate to the changes in arlr mutants (none at 1 week, reduction at 3 weeks)? Comparing Fig. 1e to 2d shows a higher relative TAG content at 1 week of both, controls and mutants compared to aged flies.

13) Line 160ff: 24h starvation-induced decrease of TAG in 1 week of age is moderate in controls (females?) as early starvation is largely based on glycogen consumption. What is the glycogen content of the mutants (compared to the controls) at this age? Depletion of (at this age) reduced glycogen stores could promote increased TAG mobilization.

14) Which part of the adult fat body was dissected for the experiments? All figure panels showing fat body LD content need careful re-inspection. Sometime nuclear sizes are very different or nuclei even not visible at all.

15) The authors are recommended to use the primary FlyBase name Lsd-1 (or plin1) not Lsd1 as LSD1

is often confused with lysine-specific demethylase.

16) Line 171: "... increase in lipid consumption..." is misleading as high and poor nutrition diet do not only differ in lipids.

17) Line 172: "fed with 10% the amount of standard food". What does this refer to? Food intake or caloric content or? Similarly in line 175 "...20% more nutrition". Please specify.

18) Line 186ff: According to Fig. S2d, arlr deltaSP-HA is expressed lower compared to arlr-HA but shows a gain-of-function phenotype rather than a rescue in Fig. 3i,j. These issues should be addressed first here and not in the following section on CG3303.

19) Line 212 should read "but not ppl>arl-HA..."

20) Line 339: This statement appears misleading as CG5162 mRNA up-regulation and reduction of LD size are correlative consequences of exercise in larval muscle. Causality ("... leading to...") is not shown to my understanding.

21) Line 357: Data on small LDs need to be shown in the supplement.

22) Line 403: "TAGL" needs attention.

23) Line 529 ff.: The cross-linking paragraph needs clarification "... reversible chemical crosslinking..." "To prevent cross-linking...?"

24) Is there a plausible reason why EndoU is named again CG3303, deviating from the FlyBase nomenclature?

25) The authors emphasize the age-dependency of arlr, which appears to decrease again in fat body of very old flies (day 70; AFCA). How does this relate to the expression profile of Arlr candidate target genes in the fat body?

26) Fig. 7f: The cartoon is informative but collectively showing Arlr and CG3303 as EndoU is confusing. For example, it is not intuitive that reduced amounts of EndoU apparently relocate to non-ER cytoplasm in the arlr mutant. Also, the profound changes in ER morphology are not reflected. Differences between wild-type and arlr mutants look rather like smooth and rough ER. This figure needs careful redesign.

27) Are the KDEL-mCherry flies unpublished? If not, please give a reference.

REVIEWER COMMENTS

Reviewer #1 (Remarks to the Author):

Sun et al. investigated the function of Arlr, a fly homolog of ENDOU, in lipid metabolism using the *Drosophila* model system. They showed a higher expression of Arlr in the *Drosophila* fat body together with an age-dependent upregulation. The authors clearly revealed that the *arlr* mutants increased lipolysis without affecting lipogenesis, reduced lipid droplet size, and shortened lifespan, which Arlr RNAi and rescue experiments further confirmed. In addition, they found that a functional paralog of *arlr*, CG3303, and human ortholog, ENDOU can functionally substitute for Arlr in the regulation of lipid metabolism. Since the physiological role of the ENDOU protein family in controlling lipid homeostasis has not been extensively studied, this work contributes to our understanding of the functional relevance of ENDOU in this context.

In the mechanism study part, the authors found that the EndoU-like domain is essential for Arlr's role in lipid metabolism. They also identified genes differentially expressed in *arlr* mutants and mRNAs associated with Arlr. Among the lipid metabolism-related genes identified were *Lsd1*, *regucalcin*, *yip2*, and *CG5162*, whose mRNAs are bound by Arlr and upregulated in *arlr* mutants. Knockdown of these genes alleviated the reduction in lipid droplet size in *arlr* mutants. Based on these findings, the authors proposed the following model: Arlr affects lipid metabolism by degrading the mRNAs of lipolysis genes.

The findings on the physiological role of Arla in lipid metabolism are solid and strongly supported by intensive phenotypic analysis. However, the primary limitation of this study is the absence of clear evidence on the molecular mechanism of Arlr. Further investigation is needed to explore the connection between Arlr's endoribonuclease activity and its role in controlling lipid homeostasis.

Major points:

1. The abstract contains an overstatement in the assertion that "Arlr affects lipid metabolism through the degradation of the mRNAs of lipolysis genes," as there is no direct evidence presented in the manuscript for the molecular mechanism of Arlr. Moreover, the proposed model in Figure 7f is oversimplified, and additional experiments are needed to clarify the precise mechanism involved.

Response 1: To demonstrate direct regulation of Arlr of the target genes, we performed luciferase reporter assays. The mRNAs of *Lsd-1*, *regucalcin*, *yip2* and *CG5162* were downregulated in the presence of Arlr (Fig. 5a). Another line of evidence is the RIP-qRT-PCR assay (we performed formaldehyde-based crosslinking RNA immunoprecipitation and RIP without formaldehyde), which

showed that the target mRNAs bound to Arlr protein. Combined with other results including the RNA levels, the genetic interaction, and the RIP-seq assay, we conclude that Arlr is required to maintain lipid homeostasis by downregulating lipolytic genes.

We have redrawn the model figure to more accurately describe our findings (Fig. 7h).

2. The gene expression analysis of wild-type and *arlr* mutant revealed 580 up-regulated genes and 384 down-regulated genes. Additionally, the RNA Immunoprecipitation experiment identified the binding of Arlr to mRNAs of 1567 genes. However, the authors of the study only examined lipid-metabolism-related genes and ultimately selected four genes as the primary targets to elucidate the molecular mechanism of Arlr. This approach may be susceptible to prejudgment or bias, and a proper in silico analysis should have been conducted to provide the rationale for focusing on specific target genes.

Response 2: We agree that there are likely more Arlr target genes, however, based on the *arlr* mutant phenotypes, we decided to focus on upregulated genes related to lipolysis. We performed a partial genetic screen using RNAi lines. RIP-qRT-PCR and RIP-seq assay further validated the binding between Arlr and target genes. We reorganized the logic and provide additional evidence such as luciferase reporter assay to show that Arlr negatively regulates the target genes.

3. Although Arlr was found to be mainly localized in the endoplasmic reticulum (ER) and to regulate ER morphology, these findings were not directly linked to Arlr's regulatory role in lipid metabolism. Given that abnormal ER networks can be observed prior to lipid droplet defects, it is necessary to investigate how Arlr, localized on the ER, regulates lipid droplet size to bridge this gap. As such, the current data cannot conclusively support the proposed model in which Arlr binds to target mRNAs on the ER and degrades them. Further investigations are warranted to elucidate the precise mechanism of Arlr's involvement in lipid metabolism regulation.

Response 3: Previous studies found that XendoU colocalizes to the ER and controls local RNA degradation on ER membranes. Loss of XendoU transforms ER tubules to ER sheets (Schwarz and Blower, 2014). Consistent with this result, Arlr colocalizes with the ER marker KDEL and loss of *arlr* affects the ER morphology. ER subcellular localization of Arlr could be an important feature for Arlr's function, however, we have no direct evidence that ER localization of Arlr is important for the regulation of lipid droplet size. We moved the localization of Arlr in the Supplementary Figure 2 and deleted its effect on ER morphology as we feel that this is beyond the scope of our story that focuses on the observation that Arlr is required to maintain lipid homeostasis by downregulating lipolytic genes. Note that we have provided

further evidence that *Arlr* negatively regulates the levels of target mRNA (Fig. 4, 5).

4. While the expression levels of four target genes were found to be upregulated in *arlr* mutants, the direct degradation of target mRNAs by *Arlr* was not validated in this study. To clarify this point, a reporter assay in an in vitro system could be performed.

Response 4: We performed luciferase reporter assays to demonstrate the degradation of target mRNAs by *Arlr*. Two vectors containing *Arlr* and a target gene, respectively, were co-transfected in HEK293T cells. The results showed that in the presence of *Arlr*, the four candidate genes were all decreased compared to the control (Fig. 5a), indicating a negative regulation of the target mRNAs.

5. In the Methods section, the authors stated that they performed RNA immunoprecipitation (IP) sequencing using formaldehyde-based crosslinking IP experiments. While formaldehyde-based crosslinking is a well-known method for investigating protein-bound RNAs, it has unique features that should be taken into account when interpreting the data. For example, formaldehyde-based crosslinking IP efficiently fixes protein-protein interactions, which can result in the indirect binding of RNAs to a target protein. Therefore, the experimental method used in this study should be clearly specified in the main text, and the results should be presented carefully along with any possible drawbacks of the method.

Response 5: Formaldehyde-based crosslinking could fix protein-protein interactions, which can result in the indirect binding of RNAs to a target protein. To prevent cross-linking and verify the target genes that directly bind to *Arlr*, we performed RIP-qRT-PCR analysis using a reagent kit without formaldehyde (RNA Immunoprecipitation [RIP] Kit, Catalog Bes5101, BersinBio, Guangzhou, China). We confirmed that *Arlr* binds to these four target genes (Fig. 5c).

6. The absence of appropriate controls in Figures 3 and 7a-e makes it difficult to clearly establish the epistatic relationships between different genotypes. Do *ppl>UAS-GFP* or *ppl>UAS-ENDOU-HA* show the same normal phenotype as the wild type or *ppl>+*? The appropriate controls provide a baseline against which the phenotypes of the experimental genotypes can be compared, allowing for a more robust interpretation of the results.

Response 6: We have added the *ppl>UAS-GFP* control in Fig. 2 and Fig. 7. LD sizes show no difference between the mutant background control and *ppl>UAS-GFP* control (Fig. 2).

Minor points:

1. Please add p-values to Supplementary Figure 8c.

Response 1: We've added p-values to Supplementary Figure 9c in the current

version.

2. The authors performed a crosslinking-IP analysis to investigate Arlr-bound RNAs, but they used the term "RNA-chromatin immunoprecipitation sequencing". This term is no longer widely accepted and has been replaced by RIP-seq or CLIP-seq.

Response 2: We now use RNA immunoprecipitation sequencing (RIP-seq) in the revised manuscript.

Reviewer #2 (Remarks to the Author)

The organelle lipid droplet plays key roles in maintaining lipid homeostasis and energy supply. The LD dynamics and metabolism is associated with many disorders, as well as aging or reproduction. Many proteins or signalings balance the LD synthesis and degradation to regulate lipid metabolism. In this manuscript, Sun et al., attempt to explore the essential functions of CG2145, an ortholog of mammalian endoribonuclease that they named arlr, on lipid metabolism. They first generated two mutant lines of arlr using CRISPR-Cas9. Arlr mutant resulted in reduction of LD size and TAG amount but not carbohydrate, which affect lifespan. Arlr influences lipid consumption but not lipogenesis. EndoU-like domain is necessary for Arlr function in LDs. Another EndoU protein CG3303 and arlr are partially redundant in regulating LD size and TAG amount. The arlr proteins localized at ER and loss of arlr affected ER morphology. The RNA-seq and RIP-seq analysis showed that lipolysis related genes increased due to the loss of arlr. Drosophila and human ENDOU proteins played conserved role in regulating lipolysis. Although the data presented are intriguing, there are several issues that should be addressed.

Major issues:

1. The FlyAtlas data showed that CG2145 also expressed in many tissues like head. In Laneve's article (reference 49), they also confirm it. So, I wonder whether RNAi of arlr in fat body have the similar effect on LD size or lifespan compared to arlr mutants. Multiple Gal4 lines should be used, such as LPP-Gal4, ppl-Gal4 or Lsp2-Gal4. In addition, whether loss of arlr in brain cells have no non-autonomous effect on LD size in fat body. Meanwhile, more LD dyes need to be examined in Fig 1b, such as BODIPY or lipidTox.

Response 1: We used two Gal4 lines, LPP-Gal4 and ppl-Gal4, to drive arlr-RNAi and found smaller LD and shorter lifespan in arlr knockdown flies (Supplementary Figure 7).

Loss of arlr in the brain cells (*elav>arlR-RNAi*) had no non-autonomous effects on LD size in the fat body.

Response Figure 1. Loss of *arlr* in the brain has no effects on LD size.

We have added BODIPY staining in an updated Supplementary Figure 5. LD defects in *arlr* mutants are similar using BODIPY and Nile Red staining.

2. During adult fly development, I ask if *arlr* is regulated by nutritional status, starvation or high nutrition. Starvation speed the lipid consumption in fat body of *arlr* compared with the control. If so, overexpression of *arlr* in fat body would antagonize lipolysis and exhibit larger LD size. Overexpression experiment should be performed.

Response 2: *arlr* expression is decreased by 24-hour starvation which is consistent with starvation speeding up lipid consumption. Surprisingly, 24-hour high fat diet and high sugar diet decreased *arlr* as well. The decrease of *arlr* by nutritional status could disrupt lipid homeostasis.

Response Figure 2. The expression of *arlr* is affected by nutrition.

Loss of *ArLr* induced upregulation of candidate genes. However, overexpression of *arlr* in the fat body did not exhibit larger LD size. *ArLr* is required but not sufficient to regulate lipid homeostasis. Under starvation conditions, LDs in *ppl>arlr* and *ppl>arlr^{ΔSP}* flies were comparable to *ppl>UAS-GFP* control flies.

Response Figure 3. Overexpression of *arlr* has no LD consumption defects under starvation conditions.

3. Mutant of *arlr* have no significant effect on larvae or young adult fly development including size and lipid storage, how about the body weight? Or whether starvation also speed lipid consumption in fat body during larval stage?

Response 3: We measured the body weight in larvae and young adults, which showed no difference between *arlr* mutants and controls (Supplementary Figure 4).

In third instar larvae, starvation could not accelerate lipid consumption as shown by Nile Red staining.

Response Figure 4. Starvation does not accelerate lipid consumption in *arlr* mutant larvae.

4. Although the authors have proved that *arlr* and CG3033 are partially functional redundancy using genetic analysis. The mechanism is still unclear. For example, does *arlr* interact with CG3033 to form a complex? Or colocalization?

Response 4: As an endoribonuclease, each protein has an endonuclease activity *in vitro* and cleaves RNA at common and distinct sites (Laneve et al., 2017). Pan-neuronal knockdown of *dendoU* showed developmental defects, such as unexpanded wings and misoriented scutellar bristles (Laneve et al., 2017) (Response Figure 7). However, neither *arlr* mutants nor pan-neuronal knockdown of *arlr* showed these defects. In our study, excessive Arlr or DendoU induced downregulation of the potential target genes (Response Figure 8). Arlr-GFP and DendoU-HA showed colocalization in the fat body (Supplementary Figure 2). We speculate that these two proteins are partially functionally redundant and compensate with each other, but do not work as a complex.

5. Since the authors found that *arlr* localized at ER using KDEL-mCherry marker, the figs are not clear. It need more ER markers and electron microscope to confirm it. Since ER is the important organelle of lipid de novo synthesis, why ER morphology change by *arlr* mutant have little effect on lipogenesis. Is it contradictory?

Response 5: We used additional ER markers (UAS-GFP.KDEL, BDSC 9898 and BDSC 9899). The pattern of KDEL-mCherry we used in the article was largely same with KDEL.GFP. Endogenous *Arlr* was localized on the ER in adipocytes (Supplementary Figure 2).

Response Figure 5. Co-staining of KDEL-mCherry and GFP.KDEL.

The tubular ER (smooth ER) is important for lipid de novo synthesis, while the sheet ER (rough ER) is the protein processing site on which ribosomes are located. Previous study found that XendoU colocalizes with ER and controls local RNA degradation on ER membranes. Loss of XendoU transforms ER tubules to ER sheets (Schwarz and Blower, 2014). Consistent with this result, *Arlr* colocalizes with the ER marker KDEL and loss of *arlr* affects the ER morphology. ER is a subcellular localization of *Arlr*, which could be an important feature for *Arlr*'s function. However, we have no direct evidence to demonstrate that *Arlr* localization on the ER is important for the regulation of lipid droplet sizes. We moved the localization of *Arlr* in the Supplementary Figure 2 and deleted its effect on ER morphology as we feel that this is beyond the scope of our story that focuses on the observation that *Arlr* is required to maintain lipid homeostasis by downregulating lipolytic genes. It would be very interesting to address how *Arlr* regulates ER morphology and whether ER morphology affects lipogenesis.

6. Similar to lipolysis, lipophagy also play important role in lipid degradation. So, whether loss of *arlr* also promote lipophagy?

Response 6: Genes in the lipophagy pathway were not affected by *Arlr* in the RNA-seq analysis. In addition, we performed *Atg8a* and *Atg5* staining and found no obvious changes between *arlr* mutants and controls. So we conclude that *Arlr* does not regulate lipolysis through lipophagy.

Response Figure 6. Atg is normal in *arlr*²⁶² mutants. a is the heatmap in the transcriptomic analysis. b–e are the staining of Atg8a and Atg5.

Reviewer #3 (Remarks to the Author)

Sun and colleagues address lipid metabolism regulation in aging using the *Drosophila* model. In this context they functionally characterize the endoribonuclease Age-related lipid regulator (Arlr) (and to some extent the paralog EndoU/CG3303). The study convincingly demonstrates the function of *arlr* in lipid storage, its implications on lifespan and the role of the gene in shaping the ER. The authors use a combination of RNA-seq and RIP-seq to screen for possible Arlr targets and test selected candidate genes involved in lipid catabolism for their capacity to revert the *arlr* mutant phenotype. Finally, they provide evidence that the human ENDOU ortholog reverts the *arlr* mutant phenotype in support of an evolutionarily conserved function of these genes in lipid metabolism.

This highly interesting study presents a comprehensive and largely solid dataset. Yet, as outlined below, some experimental additions are required to convincingly prove the conclusions of the authors in particular with respect to the identified target genes.

Major points:

1) The authors argue that lipogenesis is not affected by the absence of *arlr*, which is surprising given the severe ER phenotype of the mutants. Their

conclusion is based on mutant phenotype reversion in response to short-term feeding with HSD and HFD. However, incorporation of labelled substrates into TAG is required to make this statement solid. In addition, since LDs are produced at the ER, the localization of lipogenesis proteins will be informative. The observed reversion of the LD and TAG content phenotypes might be due to expansion of pre-existing LDs, which reflect only part of the lipogenesis capacity of the fat body. Essentially, we need to learn to what extent the observed effects on lipid metabolism are indirect effects of ER remodeling in the absence of Arlr.

Response 1: We agree with the reviewer and deleted the statement that lipogenesis is not affected by loss of Arlr in the manuscript. It is a common method to measure the overall level of TAG by glyceride assay kit (such as in Zhao et al., 2022), however, this does not address whether lipogenesis is affected. The localization of lipogenesis proteins would indeed be informative. However, we did not find suitable antibodies against lipogenesis proteins for immunostaining in *Drosophila*, such as Seipin and FIT.

The ER morphology changed in *arlr* mutants is an intriguing phenotype. Previous study found that XendoU colocalizes with ER and controls local RNA degradation on ER membranes. Loss of XendoU transforms ER tubules to ER sheets (Schwarz and Blower, 2014). In consistent with this result, Arlr colocalizes with the ER marker KDEL and loss of *arlr* affects the ER morphology. ER is a subcellular localization of Arlr, which could be an important feature for Arlr's function, but we have no direct evidence to demonstrate how Arlr is localized on the ER and regulates lipid droplet size. We would like to put the localization of Arlr in the Supplementary Figure 2 and delete its regulation on ER morphology. We hope the current data could support our conclusion that Arlr is required to maintain lipid homeostasis by downregulating lipolytic genes as we have provided further evidence to show Arlr negatively regulates the levels of target mRNA (Fig. 4, 5).

2) For the EndoU/CG3303 experiments important information and controls are missing. Apparently just one RNAi line was used of unknown knockdown efficiency. Is EndoU/CG3303 expressed in the adult fat body? In the reversion of the EndoU/CG3303 RNAi experiment by co-expression of *arlr*-HA two UAS transgenes are driven by the ppl Gal4 transgene but a specific control for this scenario is missing. Moreover, Arlr and EndoU/CG3303 have different endonucleolytic specificities (in vitro). How does go together with the observed rescue of *arlr* by EndoU/CG3303 (over)expression?

Response 2: There are two available RNAi lines, both of which are from VDRC (P{GD3933}v9916 and P{KK111894}VIE-260B). Unfortunately, we did not get the latter one due to import issues. We tested the phenotypes reported in previous studies (Laneve et al., 2017) and confirmed that it is the same line.

Response Figure 7. Knockdown of *dendoU* in the nervous system results in unexpanded wings and misoriented scutellar bristles.

There are no tools for DendoU (CG3303) staining. We drove the expression of *dendoU*-HA in the fat body and showed that the localization of *dendoU*-HA and *arlr*-GFP was quite similar (Supplementary Figure 2).

We have added *ppl-Gal4>2xUAS-GFP* control in the statistical analysis in Fig. 6h, j.

The cleavage sites of Arlr and DendoU are partially identical. Each protein has preferential cleavage sites (in vitro) (Laneve et al., 2017). These data indicate that they could have identical and different targets. We did qRT-PCR to test if the potential Arlr targets are regulated by DendoU. Overexpressing *dendoU* showed decreased expression of the four target genes.

Response Figure 8. Overexpression of *arlr* or *dendoU* shows decreased expression of the four target genes. *Lsd-1* was decreased without significance.

3) The statement in line 255. is certainly not true given that *mdy*, which encodes the major TAG synthetic enzyme is among the DEG (upregulated; although not ≥ 2) and also identified in RIP.

Response 3: The *Drosophila* ortholog of DGAT1 is *Mdy*, which functions in the formation of triglycerides. *myd* is upregulated (FC=1.39) in the RNA-seq data. Although we found no significant difference of *myd* expression by qRT-PCR analysis (data not shown), we deleted the description "genes encoding the TAG synthesis enzymes" in the manuscript.

4) The logic concerning *Lsd-1* as one of the endonucleolytic targets of Arlr is

not conclusive. It is unclear if Arlr can cleave Lsd-1 mRNA in vitro (which applies to all predicted direct targets). Nor it is clear if the increased Lsd-1 mRNA levels in *arlr* mutants translates to increased protein levels. And if so, whether increased LSD-1 would cause the small/reduced LD *arlr* mutant phenotype in adult fat body (Bi et al., report on larval fat body only). Methods and tools are published to address these question. The Lsd-1 data shown i.e. the increased LD size upon Lsd-1 RNAi in control or *arlr* mutants are fully compatible with an *arlr*-unrelated function of Lsd-1 in LD biology. Expression of *arlr* deltaSP increases LD size and TAG content. Does it reduce Lsd-1 mRNA and can these phenotypes be reverted by Lsd-1 co-overexpression?

Response 4: To show direct regulation of Arlr to the target genes, we performed luciferase reporter assays. The mRNAs of *Lsd-1*, *regucalcin*, *yip2* and *CG5162* were downregulated in the presence of Arlr (Fig. 5a).

To demonstrate if the increased Lsd-1 mRNA levels in *arlr* mutants translate to increased protein levels, we measured the endogenous Lsd-1-GFP (BDSC #94601). Lsd-1-GFP was significantly increased in *arlr* mutants.

Response Figure 9. Lsd-1-GFP is elevated in *arlr* mutants.

However, *Lsd-1* overexpressing flies showed mild LD defects in aging flies, and expressing *Lsd-1* in the *arlr* mutants did not aggravate the mutant phenotype (Fig. 4c, e). The accelerated lipid degradation in *arlr* mutants could be due to the disruption of lipid homeostasis as the increase of multiple genes is involved in this disruption of lipid homeostasis.

As the phenotype of *Lsd-1* overexpressing flies is subtle and the expression of *Lsd-1* is increased in aging flies (Response 25), we did not test the interaction with *arlr*^{ΔSP}.

5) The evidence for the remaining three predicted direct Arlr targets is even weaker, as LD size distributions are relatively subtle (though statistically significant; Fig. 6) and all individual driver and effector controls in the wt and the *arlr* background are missing. According to Fig. S4i relative TAG amount differences in males at week 5 are very pronounced between controls and *arlr* mutants. Adding data using this method for all genotypes shown in Fig. 6c would make the conclusions much stronger.

Response 5: As the LD size is relatively weak in the genetic interactions, we measured the TAG amount in all genotypes and added *ppl>UAS-GFP* (male) control to make the conclusions stronger (Fig. 4d).

6) Line 307: Fig. 7a-e lacks the control (corresponding to Fig. 1) to appreciate if expression of human ENDOU really does not change LD size and body TAG content over control levels (like *arlr*-HA but unlike *arlr* delta SP-HA) as claimed by the authors.

Moreover, if human ENDOU function in *arlr* mutants reflects evolutionary relationship, the same or at least a substantially overlapping set of target genes are expected to be regulated. Does ENDOU expression revert the up-regulation of the four proposed target genes in *arlr* mutants?

Response 6: We have added the *ppl>UAS-GFP* control in Fig. 7.

Expressing human ENDOU showed decreased expression of the target genes, similar to expressing *arlr* and *dendoU* (Response Figure 8). In the *arlr* mutant background, *Lsd-1*, *yip2*, *CG5162*, and regucalcin were all significantly reduced by expressing human ENDOU. We have shown the expression of *yip2* and *CG5162* as examples in Fig. 7g.

7) Lifespan experiments:

a) How often were they independently reproduced? Shown is one experiment each.

b) Statistics is missing (median lifespan, statistical test).

c) How is it possible that survival rates increase (see Fig. 1h and 2e shortly after day 40)?

d) Survival curves of controls on standard food (Fig. 1h) and high nutrition diet (Fig. 2e) look remarkably similar; but more rich food appears to reduce median lifespan in contrast to the mutant. How can this be explained?

e) Dietary restriction extends lifespan. Why do controls live shorter under poor nutrition diet? The discussed “unbalanced lipid homeostasis” deserves more explanation.

Response 7:

a) Lifespan experiments were independently done twice. Shown in the figures is one experiment.

b) The median lifespan is indicated in the chart and statistics has been added in the figures.

c) We checked the original data and corrected these figures.

d) Survival curves of controls on standard food and high nutrition diet are from two independent experiments. The median lifespan is 81 days on rich food vs 77 days in *arlr* mutants ($P<0.05$).

e) Dietary restriction (DR) is the reduction of food intake without malnutrition (60-70% of fully fed). In our study the poor nutrition diet contains only 10%

amount of polenta, dry yeast and white sugar compared to the amount in standard food. This starvation condition may not provide enough nutrients to keep normal lifespan, leading to shorter lifespan in control flies.

Upon DR, flies shift their metabolism toward increasing fatty-acid synthesis and breakdown, and disruption of lipid synthesis or oxidation inhibits lifespan extension upon DR (Katewa et al., 2012), indicating that lipid homeostasis is essential for lifespan extension. In *arlr* mutants lipid consumption is severely accelerated, which indicates disrupted lipid homeostasis. We have explained this in discussion.

Minor points:

8) Line 20 and elsewhere: *arlr* not *Arlr* (both italics) for the gene abbreviation.

Response 8: We've double checked the gene name and protein name in the manuscript.

9) Line 48: DGAT1 is omitted as central lipogenesis gene, despite the fact that the *Drosophila* ortholog is identified in this study as potential *Arlr* target.

Response 9: There are many proteins that regulate lipogenesis. We have added DGAT1 in the examples.

10) Line 66: "TAG synthesis enzyme perilipin"?

Response 10: Thank you for this comment. We have changed to "LD protein perilipin".

11) Line 71: Given that adult *Drosophila* HSL mutants do not show a neutral lipid phenotype this statement is questionable.

Response 11: *dHSL^{b24}* mutant larvae have larger lipid droplets and higher TAG levels than controls (Bi et al., 2012). Mature *Hsl¹* mutant adults show no difference in the size or abundance of LDs because Hsl degrades SE but not TAG (Heier et al., 2021). TAG and SE are the most common neutral lipids in LDs. Thus, our statement "in *Drosophila* the lipid storage droplet protein 1 (Lsd-1/PLIN1) stimulates lipolysis by inducing HSL activity" appears reasonable. We have added the Heier's reference in the manuscript.

12) Line 128: What does the LD size distribution and number in 1 week old controls to 3 week old controls look like? How does this relate to the changes in *arlr* mutants (none at 1 week, reduction at 3 weeks)? Comparing Fig. 1e to 2d shows a higher relative TAG content at 1 week of both, controls and mutants compared to aged flies.

Response 12: We compared the size distribution at 5 weeks in the mutants and its background control flies in order to show the decreased LD size more clearly (Fig.1d). The LD phenotype is not as obvious at 1 or 3 weeks as it is in 5-weeks old flies. The smallest LDs look more like in 1 week mutants than that

in the controls. The percentage of small LDs increased with age.

Response Figure 10. Comparison of size distribution at 1 week and 3 weeks.

Due to the development of the lipid system after eclosion, the relative TAG content at 1 week is not as steady as in older flies. We repeated three times and three replicates each time. The relative TAG is between 1.8 and 2.5. We changed the result in Fig. 2d, which is comparable to that at 3 and 5 weeks to address the confusion.

13) Line 160ff: 24h starvation-induced decrease of TAG in 1 week of age is moderate in controls (females?) as early starvation is largely based on glycogen consumption. What is the glycogen content of the mutants (compared to the controls) at this age? Depletion of (at this age) reduced glycogen stores could promote increased TAG mobilization.

Response 13: We measured the glucose content in the mutants at 1 week. It is comparable to the controls. After 24 hours starvation, glucose was decreased, which could promote increased TAG mobilization. Moreover, 5-week mutants starved for 24 hours also showed faster lipid consumption (data not shown). So we conclude that "lipolysis is increased in the absence of Arlr".

Response Figure 11. Glucose is decreased after starvation in *arlr* mutants.

14) Which part of the adult fat body was dissected for the experiments? All figure panels showing fat body LD content need careful re-inspection. Sometime nuclear sizes are very different or nuclei even not visible at all.

Response 14: Fat body of the proximal abdomen was dissected and imaged. The nuclear sizes are very different or nuclei are invisible because of the difference in focal planes. As the sample is not flat, we try to show the middle layer of the nucleus. The quantification of LD size used LDs at the middle layer of the nucleus. The nuclear size is larger in larvae than in adults.

15) The authors are recommended to use the primary FlyBase name Lsd-1 (or *plin1*) not Lsd1 as LSD1 is often confused with lysine-specific demethylase.

Response 15: We have corrected as Lsd-1.

16) Line 171: "... increase in lipid consumption..." is misleading as high and poor nutrition diet do not only differ in lipids.

Response 16: High and poor nutrition diet could change the lipid contents as well as other energies, such as sugar and protein, which may also affect the lifespan. We changed the statement to "we tested whether the decrease in lifespan of *arlr* mutants is related to reduced nutrition by feeding flies with either a low or a high nutrition diet".

17) Line 172: "fed with 10% the amount of standard food". What does this refer to? Food intake or caloric content or? Similarly in line 175 "...20% more nutrition". Please specify.

Response 17: We specified the food ingredients in Supplementary Table 3. High nutrition diet contains 20% more polenta, dry yeast and white sugar than the amount in standard food, while poor nutrition diet contains only 10% amount of the above three nutrients.

18) Line 186ff: According to Fig. S2d, *arlr* deltaSP-HA is expressed lower compared to *arlr*-HA but shows a gain-of-function phenotype rather than a rescue in Fig. 3i,j. These issues should be addressed first here and not in the following section on CG3303.

Response 18: The lower amount of *arlr*^{ΔSP}-HA than *arlr*-HA is probably due to less loading. We repeated the Western analysis and found comparable expression levels (Supplementary Figure 3d). We deleted "these defects could be rescued by expressing the protein without the signal peptide" and explained that "*ppl*>*UAS-arlr*^{ΔSP}-HA showed larger LDs. This gain-of-function phenotype suggests that the function of Arlr in the fat body is independent of its putative signal peptide and does not require secretion".

19) Line 212 should read "but not *ppl*>*arlr*-HA..."

Response 19: It should be "in *ppl>arlr-HA* flies".

20) Line 339: This statement appears misleading as CG5162 mRNA up-regulation and reduction of LD size are correlative consequences of exercise in larval muscle. Causality ("... leading to....") is not shown to my understanding.

Response 20: We have changed to "Among the targets, CG5162 mRNA levels have been previously reported to increase in response to acute exercise which leads to a significant reduction in LD size. This is consistent with its role in lipolysis".

21) Line 357: Data on small LDs need to be shown in the supplement.

Response 21: We showed the small LDs at the peripheral layer in Supplementary Figure 5g, h.

22) Line 403: "TAGL" needs attention.

Response 22: Thank you. It should be "ATGL".

23) Line 529 ff.: The cross-linking paragraph needs clarification "... reversible chemical crosslinking... "To prevent cross-linking..."?"

Response 23: We have rewritten this section as: "containing RNAs and Arlr-GFP proteins were cross-linked by 180 μ L 37% formaldehyde at room temperature and gently shaken for 15 min. To stop cross-linking, 20 μ L glycine with a final concentration of 1.25 M was mixed using a vortex for 5 min."

24) Is there a plausible reason why EndoU is named again CG3303, deviating from the FlyBase nomenclature?

Response 24: There are two endoribonucleases (CG2145 and CG3303) belonging to the eukaryotic EndoU homologs in *Drosophila*. Laneve et al. (2017) named CG3303 as *Drosophila* Endoribonuclease U-specific (DendoU) as shown in FlyBase (EndoU). CG3303 and EndoU or DendoU are the same gene products. We now changed CG3303 to DendoU in this manuscript.

25) The authors emphasize the age-dependency of arlr, which appears to decrease again in fat body of very old flies (day 70; AFCA). How does this relate to the expression profile of Arlr candidate target genes in the fat body?

Response 25: During normal aging flies, the expression of the candidate target gene *Lsd-1* is increased at 30 days, 50 days and 70 days compared to 5 days. However, it is lower at 70 days compared to 50 days, which is similar to *arlr* expression. In very old animals, many genes may decrease due to reduced metabolism.

Response Figure 12. The expression of *Lsd-1* in the fat body during aging from AFCA.

26) Fig. 7f: The cartoon is informative but collectively showing Arlr and CG3303 as EndoU is confusing. For example, it is not intuitive that reduced amounts of EndoU apparently relocate to non-ER cytoplasm in the arlr mutant. Also, the profound changes in ER morphology are not reflected. Differences between wild-type and arlr mutants look rather like smooth and rough ER. This figure needs careful redesign.

Response 26: We redesign this model figure and hope that it better represents our findings (Figure 7g).

27) Are the KDEL-mCherry flies unpublished? If not, please give a reference.

Response 27: This fly line (*KDEL-mCherry*) has not been published. It's a gift from Dr. Yixian Cui at Wuhan University. We noted "unpublished" in the article.

References:

Katwa, S. D. et al. Intramyocellular fatty-acid metabolism plays a critical role in mediating responses to dietary restriction in *Drosophila melanogaster*. *Cell Metab.* 16, 97-103 (2012).

Laneve, P. et al. *Drosophila* CG3303 is an essential endoribonuclease linked to TDP-43-mediated neurodegeneration. *Sci. Rep.* 7, 41559 (2017).

Schwarz, D. S. & Blower, M. D. The calcium-dependent ribonuclease XendoU promotes ER network formation through local RNA degradation. *J. Cell Biol.* 207, 41-57 (2014).

Zhao, X. et al. An RDH-Plin2 axis modulates lipid droplet size by antagonizing Bmm lipase. *EMBO Rep* 23, e52669 (2022).

Reviewers' Comments:

Reviewer #1:

Remarks to the Author:

The authors have addressed the critical points I raised in the previous review and significantly improved the manuscript. They have provided compelling evidence supporting the direct regulation of Arlr on target mRNAs through the use of RIP-qRT-PCR and luciferase reporter assays. Additionally, they have edited the manuscript to enhance logical deduction, making it much easier to follow.

I have a few suggestions and also need to point out an error:

The authors mentioned that both RIP-seq based on crosslinking and RIP-qRT-PCR without crosslinking were performed in this study. However, a detailed description of the crosslinking in RIP-seq and RIP-qRT-PCR experiments was only found in the Method section. It is crucial to include this information in the main text, preferably in the Results section or at least in the figure legends.

In Figure 5 legend, there is a mismatch between the descriptions and figures. Please ensure that the correct figures are paired with their respective descriptions to avoid confusion for readers.

Reviewer #2:

Remarks to the Author:

The reviewer finds that the authors addressed most concerns. Overall, this is excellent work. My opinion is that the manuscript deserves to be published.

Reviewer #3:

Remarks to the Author:

REVIEWER COMMENTS

General comment of the reviewer 3:

The substantially revised version of the study by Sun and colleagues represents a careful, detailed and major improvement. Many of my comments and concerns were appropriately addressed (exceptions see below). Some of new data are already part of the revised version; some other data made accessible to this reviewer also future readers would benefit from if they were represented in the supplement (see below).

The authors did the best possible to close the gap between some few selected direct targets of Arlr and their involvement in the (aging) lipid homeostasis phenotype of arlr mutants. The pleiotropic effects of Arlr might prevent that this gap becomes fully closed.

Reviewer #3 (Remarks to the Author)

Sun and colleagues address lipid metabolism regulation in aging using the *Drosophila* model. In this context they functionally characterize the endoribonuclease Age-related lipid regulator (Arlr) (and to some extent the paralog EndoU/CG3303). The study convincingly demonstrates the function of arlr in lipid storage, its implications on lifespan and the role of the gene in shaping the ER. The authors use a combination of RNA-seq and RIP-seq to screen for possible Arlr targets and test selected candidate genes involved in lipid catabolism for their capacity to revert the arlr mutant phenotype. Finally, they provide evidence that the human ENDOU ortholog reverts the arlr mutant phenotype in support of an evolutionarily conserved function of these genes in lipid metabolism.

This highly interesting study presents a comprehensive and largely solid dataset. Yet, as outlined below, some experimental additions are required to convincingly prove the conclusions of the authors in particular with respect to the identified target genes.

Major points:

2) For the EndoU/CG3303 experiments important information and controls are missing. Apparently just one RNAi line was used of unknown knockdown efficiency. Is EndoU/CG3303 expressed in the adult fat body? In the reversion of the EndoU/CG3303 RNAi experiment by co-expression of arlr-HA two UAS transgenes are driven by the ppl Gal4 transgene but a specific control for this scenario is missing. Moreover, Arlr and EndoU/CG3303 have different endonucleolytic specificities (in vitro). How does go together with the observed rescue of arlr by EndoU/CG3303 (over)expression?

Response 2: There are two available RNAi lines, both of which are from VDRC (P{GD3933}v9916 and P{KK111894}VIE-260B). Unfortunately, we did not get the latter one due to import issues.

Reply of the reviewer: The authors have my sympathy for being affected by restrictions to the free exchange of research material. On the other hand, this is no good argument to provide independent experimental prove. First, it sounds odd that one bot not a second fly line could be acquired from VDRC. Second, being a co-author of the study Dr Perrimon at Harvard is unlikely be affected by fly stock import issues.

We tested the phenotypes reported in previous studies (Laneve et al., 2017) and confirmed that it is the same line.

Response Figure 7. Knockdown of dendoU in the nervous system results in unexpanded wings and misoriented scutellar bristles.

There are no tools for DendoU (CG3303) staining. We drove the expression of dendoU-HA in the fat body and showed that the localization of dendoU-HA and arlr-GFP was quite similar (Supplementary Figure 2).

We have added ppl-Gal4>2xUAS-GFP control in the statistical analysis in Fig. 6h, j.

Reply of the reviewer: Thank you.

The cleavage sites of Arlr and DendoU are partially identical. Each protein has preferential cleavage sites (in vitro) (Laneve et al., 2017). These data indicate that they could have identical and different targets. We did qRT-PCR to test if the potential Arlr targets are regulated by DendoU. Overexpressing dendoU showed decreased expression of the four target genes.

Response Figure 8. Overexpression of arlr or dendoU shows decreased expression of the four target genes. Lsd-1 was decreased without significance.

Reply of the reviewer: The authors appear to argue now that with respect to intracellular localization of the proteins and just looking at these four potential direct target genes the tow paralogs behave identically and therefore fully redundant. Is this correct?

3) The statement in line 255. is certainly not true given that mdy, which encodes the major TAG synthetic enzyme is among the DEG (upregulated; although not ≥ 2) and also identified in RIP.

Response 3: The Drosophila ortholog of DGAT1 is Mdy, which functions in the formation of triglycerides. myd is upregulated (FC=1.39) in the RNA-seq data. Although we found no significant difference of myd expression by qRT-PCR analysis (data not shown), we deleted the description "genes encoding the TAG synthesis enzymes" in the manuscript.

Reply of the reviewer: Given that mdy/DGAT1 is among the "Lipid metabolism-related DEGs" according to the data in the manuscript the statement is misleading: "In contrast, genes involved in LD biogenesis and growth,... were not found among the DEGs (Supplementary Table 1)."

4) The logic concerning Lsd-1 as one of the endonucleolytic targets of Arlr is not conclusive. It is unclear if Arlr can cleave Lsd-1 mRNA in vitro (which applies to all predicted direct targets). Nor it is clear if the increased Lsd-1 mRNA levels in arlr mutants translates to increased protein levels. And if so, whether increased LSD-1 would cause the small/reduced LD arlr mutant phenotype in adult fat

body (Bi et al., report on larval fat body only). Methods and tools are published to address these question. The Lsd-1 data shown i.e. the increased LD size upon Lsd-1 RNAi in control or arlr mutants are fully compatible with an arlr-unrelated function of Lsd-1 in LD biology. Expression of arlr deltaSP increases LD size and TAG content. Does it reduce Lsd-1 mRNA and can these phenotypes be reverted by Lsd-1 co-overexpression?

Response 4: To show direct regulation of Arlr to the target genes, we performed luciferase reporter assays. The mRNAs of Lsd-1, regucalcin, yip2 and CG5162 were downregulated in the presence of Arlr (Fig. 5a).

To demonstrate if the increased Lsd-1 mRNA levels in arlr mutants translate to increased protein levels, we measured the endogenous Lsd-1-GFP (BDSC #94601). Lsd-1-GFP was significantly increased in arlr mutants.

Response Figure 9. Lsd-1-GFP is elevated in arlr mutants.

However, Lsd-1 overexpressing flies showed mild LD defects in aging flies, and expressing Lsd-1 in the arlr mutants did not aggravate the mutant phenotype (Fig. 4c, e). The accelerated lipid degradation in arlr mutants could be due to the disruption of lipid homeostasis as the increase of multiple genes is involved in this disruption of lipid homeostasis.

As the phenotype of Lsd-1 overexpressing flies is subtle and the expression of Lsd-1 is increased in aging flies (Response 25), we did not test the interaction with arlr Δ SP.

Reply of the reviewer: The authors are very much acknowledged for their extra experimental efforts. Apparently, the contribution of the LSD-1 regulation by Arlr is very clear (Response figure 9 should be in the supplement of make this clear to the reader) but the contribution of (increased) LSD-1 on the aging lipid homeostasis is very limited.

5) The evidence for the remaining three predicted direct Arlr targets is even weaker, as LD size distributions are relatively subtle (though statistically significant; Fig. 6) and all individual driver and effector controls in the wt and the arlr background are missing. According to Fig. S4i relative TAG amount differences in males at week 5 are very pronounced between controls and arlr mutants. Adding data using this method for all genotypes shown in Fig. 6c would make the conclusions much stronger.

Response 5: As the LD size is relatively weak in the genetic interactions, we measured the TAG amount in all genotypes and added ppl>UAS-GFP (male) control to make the conclusions stronger (Fig. 4d).

Reply of the reviewer: This makes the point much stronger.

7) Lifespan experiments:

- a) How often were they independently reproduced? Shown is one experiment each.
- b) Statistics is missing (median lifespan, statistical test).
- c) How is it possible that survival rates increase (see Fig. 1h and 2e shortly after day 40)?
- d) Survival curves of controls on standard food (Fig. 1h) and high nutrition diet (Fig. 2e) look remarkably similar; but more rich food appears to reduce median lifespan in contrast to the mutant. How can this be explained?
- e) Dietary restriction extends lifespan. Why do controls live shorter under poor nutrition diet? The discussed "unbalanced lipid homeostasis" deserves more explanation.

Response 7:

- a) Lifespan experiments were independently done twice. Shown in the figures is one experiment.

Reply of the reviewer:

Please add the independent lifespan data to the supplement.

- b) The median lifespan is indicated in the chart and statistics has been added in the figures.

Reply of the reviewer: Figure 3 legend reads "The lifespan of arlr mutants is restored by high nutrition diet in females. The median lifespan is indicated in the chart. Statistical data were analyzed by independent two-sample t tests. This statistical analysis is inappropriate for lifespan curves and needs improvement.

Minor points:

11) Line 71: Given that adult *Drosophila* HSL mutants do not show a neutral lipid phenotype this statement is questionable.

Response 11: dHSLb24 mutant larvae have larger lipid droplets and higher TAG levels than controls (Bi et al., 2012). Mature Hsl1 mutant adults show no difference in the size or abundance of LDs because Hsl degrades SE but not TAG (Heier et al., 2021). TAG and SE are the most common neutral lipids in LDs. Thus, our statement "in *Drosophila* the lipid storage droplet protein 1 (Lsd-1/PLIN1) stimulates lipolysis by inducing HSL activity" appears reasonable. We have added the Heier's reference in the manuscript.

Reply of the reviewer: I respectfully disagree with the authors. In adult flies - the stage relevant for the presented study - there is so far at least to my knowledge no experimental support for LSD-1 stimulating HSL activity. This is based on Lsd-1 mutants but not Hsl mutants having a TAG accumulation phenotype.

12) Line 128: What does the LD size distribution and number in 1 week old controls to 3 week old controls look like? How does this relate to the changes in arlr mutants (none at 1 week, reduction at 3 weeks)? Comparing Fig. 1e to 2d shows a higher relative TAG content at 1 week of both, controls and mutants compared to aged flies.

Response 12: We compared the size distribution at 5 weeks in the mutants and its background control flies in order to show the decreased LD size more clearly (Fig.1d). The LD phenotype is not as obvious at 1 or 3 weeks as it is in 5-weeks old flies. The smallest LDs look more like in 1 week mutants than that in the controls. The percentage of small LDs increased with age.

Response Figure 10. Comparison of size distribution at 1 week and 3 weeks.

Due to the development of the lipid system after eclosion, the relative TAG content at 1 week is not as steady as in older flies. We repeated three times and three replicates each time. The relative TAG is between 1.8 and 2.5. We changed the result in Fig. 2d, which is comparable to that at 3 and 5 weeks to address the confusion.

Reply of the reviewer: Thank you for this clarification.

REVIEWERS' COMMENTS

Reviewer #1 (Remarks to the Author):

The authors have addressed the critical points I raised in the previous review and significantly improved the manuscript. They have provided compelling evidence supporting the direct regulation of Arlr on target mRNAs through the use of RIP-qRT-PCR and luciferase reporter assays. Additionally, they have edited the manuscript to enhance logical deduction, making it much easier to follow.

I have a few suggestions and also need to point out an error:

The authors mentioned that both RIP-seq based on crosslinking and RIP-qRT-PCR without crosslinking were performed in this study. However, a detailed description of the crosslinking in RIP-seq and RIP-qRT-PCR experiments was only found in the Method section. It is crucial to include this information in the main text, preferably in the Results section or at least in the figure legends.

Response to the reviewer: We have now included a detailed description of the crosslinking for the RIP-seq and RIP-qRT-PCR experiments in the main text.

In Figure 5 legend, there is a mismatch between the descriptions and figures. Please ensure that the correct figures are paired with their respective descriptions to avoid confusion for readers.

Response to the reviewer: Thank you for the correction. We have corrected the figure description.

Reviewer #2 (Remarks to the Author):

The reviewer finds that the authors addressed most concerns. Overall, this is excellent work. My opinion is that the manuscript deserves to be published.

Response to the reviewer: Thank you for the comments.

Reviewer #3 (Remarks to the Author):

REVIEWER COMMENTS

General comment of the reviewer 3:

The substantially revised version of the study by Sun and colleagues represents a careful, detailed and major improvement. Many of my comments and concerns were appropriately addressed (exceptions see below). Some of new data are already part of the revised version; some other data made accessible to this reviewer also future readers would benefit from if they were represented in the supplement (see below).

The authors did the best possible to close the gap between some few selected direct targets of Arlr and their involvement in the (aging) lipid homeostasis

phenotype of *arlr* mutants. The pleiotropic effects of *Arlr* might prevent that this gap becomes fully closed.

Reviewer #3 (Remarks to the Author)

Sun and colleagues address lipid metabolism regulation in aging using the *Drosophila* model. In this context they functionally characterize the endoribonuclease Age-related lipid regulator (*Arlr*) (and to some extent the paralog *EndoU/CG3303*). The study convincingly demonstrates the function of *arlr* in lipid storage, its implications on lifespan and the role of the gene in shaping the ER. The authors use a combination of RNA-seq and RIP-seq to screen for possible *Arlr* targets and test selected candidate genes involved in lipid catabolism for their capacity to revert the *arlr* mutant phenotype. Finally, they provide evidence that the human *ENDO U* ortholog reverts the *arlr* mutant phenotype in support of an evolutionarily conserved function of these genes in lipid metabolism.

This highly interesting study presents a comprehensive and largely solid dataset. Yet, as outlined below, some experimental additions are required to convincingly prove the conclusions of the authors in particular with respect to the identified target genes.

Major points:

2) For the *EndoU/CG3303* experiments important information and controls are missing. Apparently just one RNAi line was used of unknown knockdown efficiency. Is *EndoU/CG3303* expressed in the adult fat body? In the reversion of the *EndoU/CG3303* RNAi experiment by co-expression of *arlr*-HA two UAS transgenes are driven by the *ppl Gal4* transgene but a specific control for this scenario is missing. Moreover, *Arlr* and *EndoU/CG3303* have different endonucleolytic specificities (in vitro). How does go together with the observed rescue of *arlr* by *EndoU/CG3303* (over)expression?

Response 2: There are two available RNAi lines, both of which are from VDRC (*P{GD3933}v9916* and *P{KK111894}VIE-260B*). Unfortunately, we did not get the latter one due to import issues.

Reply of the reviewer: The authors have my sympathy for being affected by restrictions to the free exchange of research material. On the other hand, this is no good argument to provide independent experimental prove. First, it sounds odd that one bot not a second fly line could be acquired from VDRC. Second, being a co-author of the study Dr Perrimon at Harvard is unlikely be affected by fly stock import issues.

Response to the reply of the reviewer: We conducted the experiments in China where we can order VDRC lines in March-April or August-September (because of the outside temperature) through a company. However, we missed the time window and we cannot order the flies on our own without an import permission. We could have constructed an independent line as a backup, however, we regretfully did not, due to time limitation. We have added a sentence: "Note

that the experiment with *dendoU* RNAi was performed with a single RNAi line so the conclusion may need to be confirmed with additional lines in the future.”

We tested the phenotypes reported in previous studies (Laneve et al., 2017) and confirmed that it is the same line.

Response Figure 7. Knockdown of *dendoU* in the nervous system results in unexpanded wings and misoriented scutellar bristles.

There are no tools for *DendoU* (CG3303) staining. We drove the expression of *dendoU*-HA in the fat body and showed that the localization of *dendoU*-HA and *arlr*-GFP was quite similar (Supplementary Figure 2).

We have added *ppl-Gal4>2xUAS-GFP* control in the statistical analysis in Fig. 6h, j.

Reply of the reviewer: Thank you.

The cleavage sites of *Arlr* and *DendoU* are partially identical. Each protein has preferential cleavage sites (in vitro) (Laneve et al., 2017). These data indicate that they could have identical and different targets. We did qRT-PCR to test if the potential *Arlr* targets are regulated by *DendoU*. Overexpressing *dendoU* showed decreased expression of the four target genes.

Response Figure 8. Overexpression of *arlr* or *dendoU* shows decreased expression of the four target genes. *Lsd-1* was decreased without significance.

Reply of the reviewer: The authors appear to argue now that with respect to intracellular localization of the proteins and just looking at these four potential direct target genes the two paralogs behave identically and therefore fully redundant. Is this correct?

Response to the reply of the reviewer: Although their intracellular localization is similar and *DendoU* could regulate the expression of the potential target genes, we did not conclude that the two paralogs behave identically since we did not test whether *DendoU* binds to these mRNAs and the LD phenotype of *dendoU* and *arlr* mutants is not identical. Thus, we do not know whether they are fully redundant. Our conclusion is that "our findings reveal a conserved role of *EndoU* family proteins in lipid metabolism and that these proteins may function through distinct targets but have complementary roles in lipid accumulation".

3) The statement in line 255. is certainly not true given that *mdy*, which encodes the major TAG synthetic enzyme is among the DEG (upregulated; although not ≥ 2) and also identified in RIP.

Response 3: The *Drosophila* ortholog of DGAT1 is *Mdy*, which functions in the formation of triglycerides. *myd* is upregulated (FC=1.39) in the RNA-seq data. Although we found no significant difference of *myd* expression by qRT-PCR analysis (data not shown), we deleted the description "genes encoding the TAG synthesis enzymes" in the manuscript.

Reply of the reviewer: Given that *mdy*/DGAT1 is among the "Lipid metabolism-related DEGs" according to the data in the manuscript the

statement is misleading: "In contrast, genes involved in LD biogenesis and growth,... were not found among the DEGs (Supplementary Table 1)."

Response to the reply of the reviewer: We modified the statement to "few genes involved in LD biogenesis and growth, as well as autophagy, were found among the DEGs".

4) The logic concerning Lsd-1 as one of the endonucleolytic targets of Arlr is not conclusive. It is unclear if Arlr can cleave Lsd-1 mRNA in vitro (which applies to all predicted direct targets). Nor it is clear if the increased Lsd-1 mRNA levels in arlr mutants translates to increased protein levels. And if so, whether increased LSD-1 would cause the small/reduced LD arlr mutant phenotype in adult fat body (Bi et al., report on larval fat body only). Methods and tools are published to address these question. The Lsd-1 data shown i.e. the increased LD size upon Lsd-1 RNAi in control or arlr mutants are fully compatible with an arlr-unrelated function of Lsd-1 in LD biology. Expression of arlr deltaSP increases LD size and TAG content. Does it reduce Lsd-1 mRNA and can these phenotypes be reverted by Lsd-1 co-overexpression?

Response 4: To show direct regulation of Arlr to the target genes, we performed luciferase reporter assays. The mRNAs of Lsd-1, regucalcin, yip2 and CG5162 were downregulated in the presence of Arlr (Fig. 5a).

To demonstrate if the increased Lsd-1 mRNA levels in arlr mutants translate to increased protein levels, we measured the endogenous Lsd-1-GFP (BDSC #94601). Lsd-1-GFP was significantly increased in arlr mutants.

Response Figure 9. Lsd-1-GFP is elevated in arlr mutants.

However, Lsd-1 overexpressing flies showed mild LD defects in aging flies, and expressing Lsd-1 in the arlr mutants did not aggravate the mutant phenotype (Fig. 4c, e). The accelerated lipid degradation in arlr mutants could be due to the disruption of lipid homeostasis as the increase of multiple genes is involved in this disruption of lipid homeostasis.

As the phenotype of Lsd-1 overexpressing flies is subtle and the expression of Lsd-1 is increased in aging flies (Response 25), we did not test the interaction with arlr Δ SP.

Reply of the reviewer: The authors are very much acknowledged for their extra experimental efforts. Apparently, the contribution of the LSD-1 regulation by Arlr is very clear (Response figure 9 should be in the supplement of make this clear to the reader) but the contribution of (increased) LSD-1 on the aging lipid homeostasis is very limited.

Response to the reply of the reviewer: None of these target genes we found show age-dependent regulation by Arlr. Overexpression of a single gene such as *Lsd-1* has subtle LD defects during aging. We agree that the contribution of a single gene on the aging lipid homeostasis is limited. There is probably an essential target gene involved in lipid homeostasis during aging. We hope to identify it in the future.

We prefer not to show the Lsd-1-GFP staining (in Response figure 9) in the

supplementary figures because we do not want to highlight the role of Lsd-1 out of the four targets.

5) The evidence for the remaining three predicted direct Arlr targets is even weaker, as LD size distributions are relatively subtle (though statistically significant; Fig. 6) and all individual driver and effector controls in the wt and the arlr background are missing. According to Fig. S4i relative TAG amount differences in males at week 5 are very pronounced between controls and arlr mutants. Adding data using this method for all genotypes shown in Fig. 6c would make the conclusions much stronger.

Response 5: As the LD size is relatively weak in the genetic interactions, we measured the TAG amount in all genotypes and added ppl>UAS-GFP (male) control to make the conclusions stronger (Fig. 4d).

Reply of the reviewer: This makes the point much stronger.

7) Lifespan experiments:

a) How often were they independently reproduced? Shown is one experiment each.

b) Statistics is missing (median lifespan, statistical test).

c) How is it possible that survival rates increase (see Fig. 1h and 2e shortly after day 40)?

d) Survival curves of controls on standard food (Fig. 1h) and high nutrition diet (Fig. 2e) look remarkably similar; but more rich food appears to reduce median lifespan in contrast to the mutant. How can this be explained?

e) Dietary restriction extends lifespan. Why do controls live shorter under poor nutrition diet? The discussed “unbalanced lipid homeostasis” deserves more explanation.

Response 7:

a) Lifespan experiments were independently done twice. Shown in the figures is one experiment.

Reply of the reviewer: Please add the independent lifespan data to the supplement.

Response to the reply of the reviewer: We show the independent lifespan data of all figures below. a is an independent repeat of Fig. 1h, b is a repeat of Fig. 3e, c is for Supplementary Fig. 6l, d is for Supplementary Fig. 7d, and e is for Supplementary Fig. 7h. The lifespan and median lifespan in two replicates are similar. All lifespan tests are replicable in this study. There are five lifespan figures in total in this study. If we add all replicated data to the supplementary information and explain each one in the text, we are afraid that the article would be wordy and redundant. We explained in Methods that “Lifespan test was repeatable for each genotype and one of the two repeats was shown in the figures”.

b) The median lifespan is indicated in the chart and statistics has been added in the figures.

Reply of the reviewer: Figure 3 legend reads “The lifespan of arlr mutants is restored by high nutrition diet in females. The median lifespan is indicated in the chart. Statistical data were analyzed by independent two-sample t tests. This statistical analysis is inappropriate for lifespan curves and needs improvement.

Response to the reply of the reviewer: The statistical analysis of lifespan is by Log-rank (Mantel-Cox) test. We have corrected the figure legends.

Minor points:

11) Line 71: Given that adult *Drosophila* HSL mutants do not show a neutral lipid phenotype this statement is questionable.

Response 11: dHSLb24 mutant larvae have larger lipid droplets and higher TAG levels than controls (Bi et al., 2012). Mature Hsl1 mutant adults show no difference in the size or abundance of LDs because Hsl degrades SE but not TAG (Heier et al., 2021). TAG and SE are the most common neutral lipids in LDs. Thus, our statement “in *Drosophila* the lipid storage droplet protein 1 (Lsd-1/PLIN1) stimulates lipolysis by inducing HSL activity” appears reasonable. We have added the Heier’s reference in the manuscript.

Reply of the reviewer: I respectfully disagree with the authors. In adult flies - the stage relevant for the presented study - there is so far at least to my knowledge no experimental support for LSD-1 stimulating HSL activity. This is based on Lsd-1 mutants but not Hsl mutants having a TAG accumulation phenotype.

Response to the reply of the reviewer: Lsd-1 does not stimulate HSL activity directly, but by recruiting the protein to the LD surface. We changed to “in *Drosophila* the lipid storage droplet protein 1 (Lsd-1/PLIN1) stimulates lipolysis by recruiting HSL to the LD surface”.

12) Line 128: What does the LD size distribution and number in 1 week old controls to 3 week old controls look like? How does this relate to the changes in arlr mutants (none at 1 week, reduction at 3 weeks)? Comparing Fig. 1e to 2d shows a higher relative TAG content at 1 week of both, controls and mutants compared to aged flies.

Response 12: We compared the size distribution at 5 weeks in the mutants and its background control flies in order to show the decreased LD size more clearly (Fig.1d). The LD phenotype is not as obvious at 1 or 3 weeks as it is in 5-weeks old flies. The smallest LDs look more like in 1 week mutants than that in the controls. The percentage of small LDs increased with age.

Response Figure 10. Comparison of size distribution at 1 week and 3 weeks. Due to the development of the lipid system after eclosion, the relative TAG content at 1 week is not as steady as in older flies. We repeated three times and three replicates each time. The relative TAG is between 1.8 and 2.5. We changed the result in Fig. 2d, which is comparable to that at 3 and 5 weeks to address the confusion.

Reply of the reviewer: Thank you for this clarification.